# Aircraft Emissions, Their Plume-Scale Effects, and the Spatio-Temporal Sensitivity of the Atmospheric Response: A Review

Kieran N. Tait [1],*, Mohammad Anwar H. Khan [2], Steve Bullock [1], Mark H. Lowenberg [1] and Dudley E. Shallcross [2]

1   Department of Aerospace Engineering, Queen's Building, University Walk, University of Bristol, Bristol BS8 1TR, UK; steve.bullock@bristol.ac.uk (S.B.); m.lowenberg@bristol.ac.uk (M.H.L.)
2   Atmospheric Chemistry Research Group, School of Chemistry, Cantock's Close, University of Bristol, Bristol BS8 1TS, UK; anwar.khan@bristol.ac.uk (M.A.H.K.); d.e.shallcross@bristol.ac.uk (D.E.S.)
*   Correspondence: kieran.tait@bristol.ac.uk; Tel.: +44-7932003344

**Abstract:** Non-$CO_2$ aircraft emissions are responsible for the majority of aviation's climate impact, however their precise effect is largely dependent on the environmental conditions of the ambient air in which they are released. Investigating the principal causes of this spatio-temporal sensitivity can bolster understanding of aviation-induced climate change, as well as offer potential mitigation solutions that can be implemented in the interim to low carbon flight regimes. This review paper covers the generation of emissions and their characteristic dispersion, air traffic distribution, local and global climate impact, and operational mitigation solutions, all aimed at improving scientific awareness of aviation's non-$CO_2$ climate impact.

**Keywords:** aviation; aircraft emissions; climate impact; non-$CO_2$; operational mitigation; plume-scale; air traffic; climate-optimal routing; formation flight

## 1. Introduction

Aircraft act as high-altitude emissions vectors, transporting a number of radiatively and chemically active substances across vast regions of the globe. These substances induce a net global warming effect that constitutes 3.5% of global climate change due to anthropogenic emissions [1]. Whilst carbon dioxide ($CO_2$) emissions are often portrayed as the leading contributor to aviation-induced climate change, they are only responsible for one-third of aviation's net climate impact. The remaining two-thirds of this impact are attributed to reactive non-$CO_2$ emissions, primarily nitrogen oxides ($NO_x$), water vapour ($H_2O$) and particulate matter (PM). These emission species interact with ambient air through chemical and microphysical processing, giving rise to the production and depletion of radiatively active substances, which perturb the net energy balance of the atmosphere (e.g., $NO_x$-induced ozone production, condensation trail (contrail) generation through $H_2O$ and PM emissions etc.). Due to the reactive nature of non-$CO_2$ aircraft emissions, the climatic response varies depending on the state of the background atmosphere (i.e., its chemical composition and meteorology) and the time of day and year on which emissions are released. This means that aviation climate impact is spatio-temporally sensitive, i.e., the same emissions released at different times and/or locations can lead to very different atmospheric effects.

The dispersion of aircraft emissions occurs over great distance and time scales, with emissions entrained in the aircraft exhaust plume, which spreads hundreds of kilometres over its lifetime of up to 12 h [2,3]. The elevated concentrations of emitted chemical species present within the plume result in additional nonlinear chemical (gas-phase and heterogeneous) and microphysical processing, which is often not accounted for in global chemistry models, due to the inherent assumption of instantaneous dispersion (ID) of emissions.

The ID approach assumes that, immediately following release into the atmosphere, emissions are homogeneously mixed into the volume of the computational grid cell to which they are released. This means that any subgrid-scale nonlinear processing that may occur throughout the plume lifetime is often neglected [4], and hence plume-scale modelling and parametrisation is essential in high-fidelity aviation climate modelling. Furthermore, plume-scale nonlinear effects are also further augmented in high-density airspace regions, where plumes intersect and the emissions contained within them accumulate, leading to chemical saturation. Two key saturation effects have been noted to be of particular significance with respect to aviation climate impact: (1) the saturation of $NO_x$ emissions, which leads to decreasing ozone production with increasing $NO_x$ concentrations [5], and (2) the dehydration of water vapour leading to diminished contrail climate impact [6]. Therefore, the atmospheric response to non-$CO_2$ aircraft emissions is not only sensitive to natural variations in the atmospheric state, but is also affected by emissions present in lingering plumes from aircraft that have previously flown through.

Optimising flight routes with respect to minimum climate impact (as opposed to minimum fuel burn) is a concept that has become prevalent in the literature in recent years. This involves re-routing aircraft to avoid particularly climate-sensitive regions of the atmosphere, based on provision of en-route chemical and meteorological information. Niklaß et al. [7] have shown through simulation efforts that it is possible to achieve a 12% reduction in climate impact at virtually no additional cost to fuel burn, evidencing the practicability of climate-optimised routing. Other studies approach this issue purely from the perspective of superimposing aircraft plumes through formation flight. The aerodynamic benefits of formation flight due to wake energy retrieval are already well established [8,9], however it has become increasingly evident that formation flight can also be used to exploit climate-beneficial saturation effects due to the superposition of plumes from aircraft involved. Dahlmann et al. [10] explores the potential for climate impact reduction due to saturation effects in formation flight, finding that $NO_x$ saturation can reduced ozone production efficiency of ~5% and mutual inhibition of contrail growth can quell contrail radiative properties by 20 to 60%.

This review article documents the current state of literature on aviation's atmospheric effects and the influence of nonlinear plume-scale processing on the global net climate impact, to bring to light the notion that aviation-induced perturbations to atmospheric chemistry can simply be viewed as another constraint in the climate-optimised routing problem. Section 2 covers the emissions generation process and the methods used to model aircraft performance and emissions. Section 3 explores the dispersion characteristics of aircraft emissions on the subgrid scale, alluding to the various modelling methods developed to investigate the dynamical characteristics of aircraft plumes. In Section 4, air traffic management principles and their effect on aviation emissions distribution are considered, on both a local and global scale. Section 5 explores global aviation climate impact, based on the emissions released and the chemical and physical processes that occur on the grid and subgrid-scale of global models. Section 6 looks into the effect of subgrid-scale processes that occur due to emissions accumulation in high-density airspace, and finally Section 7 explores potential mitigation efforts to reduce aviation-induced climate change by modifying aircraft operations, such as climate-optimal routing and formation flight.

## 2. Aircraft Emissions

Aircraft propulsion systems provide the essential driving force component to enable powerful and efficient flight. In modern commercial aviation, responsible for ~88% of global aviation fuel usage [11], it is commonplace to use a gas turbine propulsion system operating on kerosene-based jet fuel. Aviation's environmental impact stems from the emission and dispersion of chemically and radiatively active substances that are generated during jet fuel combustion. This section details the emissions generation process and the modelling methods implemented to estimate emissions, based on aircraft performance calculations and the determination of fuel consumption throughout flight.

### 2.1. The Generation of Aircraft Emissions

The widespread usage of kerosene-based jet fuels stems from the need to satisfy strict power-to-weight and safety requirements demanded by commercial aircraft; this is because kerosene has an exceptionally high energy density and wide operating temperature range at a low financial cost, compared with alternative fuel types [12,13]. The stoichiometric combustion of kerosene with air, required to generate thrust, does however generate a number of chemically- and radiatively-active emission species which are emitted from the aircraft into its wake. These aircraft emissions interact with the surrounding atmosphere, perturbing the natural balance of chemistry and contributing to air quality issues and anthropogenic climate change. The mass of emission produced per unit mass of fuel burnt is often referred to as the emission index (EI), measured in grams of emission per kilogram of fuel [g/kg].

#### 2.1.1. Primary Jet Fuel Combustion Products

The products of jet fuel combustion are often divided into two categories: primary and secondary, alluding to the way in which they are formed inside the aircraft engine. The primary products of jet fuel combustion are $CO_2$, $H_2O$ and, due to fuel impurities, a relatively small amount of sulphur oxides ($SO_x$). These products are generated as a direct result of the combustion reaction that takes place, and hence depends on the carbon-hydrogen-sulphur composition of the fuel. The direct coupling of the production of these species to fuel consumption means that they have a constant EI throughout all phases of flight. Example values based on the typical chemical composition of jet fuel are given in Table 1 [14].

**Table 1.** Emission indices estimates for $CO_2$, $H_2O$ and $SO_2$, averaged from a range of existing studies testing various aviation fuel types. Data derived from [15].

| $CO_2$ | $H_2O$ | $SO_2$ |
|:---:|:---:|:---:|
| 3.149 g/kg fuel | 1.230 g/kg fuel | 0.84 g/kg fuel |

#### 2.1.2. Secondary Jet Fuel Combustion Products

Further to this, a number of secondary combustion products are generated in the aircraft exhaust, namely oxides of nitrogen ($NO_x \equiv NO$ & $NO_2$), carbon monoxide (CO), unburnt hydrocarbons (HC), particulate matter (PM) and trace levels of volatile organic compounds (VOCs) [16]. These products are termed secondary because their production levels differ, depending on the nature of the combustion process and the engine load condition [17]. This means that their EIs are variable throughout flight, depending on aircraft engine type, engine operating conditions, and the atmospheric conditions of the surrounding environment [18].

$NO_x$ emissions originate from the entry of atmospheric nitrogen ($N_2$) into the high temperature combustion chamber. The level of $NO_x$ increases with increasing temperature and pressure, as it is coupled to the thermal reaction processes that occur in the primary combustion zone. Therefore, with the assumption of constant polytropic and combustion efficiencies, the emission index of $NO_x$ ($EINO_x$) can be correlated with aircraft fuel flow [17,19]. As a result of inefficiencies in the combustion process, products such as CO and HC are formed. In contrast to $NO_x$, these emissions are direct products of incomplete combustion, meaning their concentrations are inversely proportional to combustion efficiency. Since combustion efficiency correlates with thrust for sea level static (SLS) conditions, and thrust correlates with fuel flow, this means that EICO and EIHC decrease with increasing fuel flow.

PM emissions from aircraft can be categorised by their volatility: non-volatile PM (nvPM) and volatile PM (vPM). The primary source of nvPM present in aircraft exhaust is soot, which constitutes the greatest warming effect of all particles released from aircraft [20]. Soot is generated in the fuel-rich regions of the combustor, where the condensation of un-

burnt aromatic hydrocarbons takes place, converting low carbon content HC fuel molecules into carbonaceous agglomerates containing millions of carbon atoms [21,22]. The extent of soot formation therefore depends on the fuel-air-ratio and the mixing processes that take place in the combustor, which vary with combustor design and are influenced by the non-homogeneous flow and temperature fields. Accurate measurements of these parameters are rare, making it very difficult to directly acquire quantitative data on soot emissions. Instead, soot EI can be estimated based on correlation with the so-called smoke number measured in engine certification procedures [19].

The formation of vPM from aircraft largely derives from the emission of sulphur derivatives, lubrication oil and VOCs [23]. During combustion, the sulphur content in the fuel is mostly oxidised to sulphur dioxide ($SO_2$), some of which is then further oxidised to sulphuric acid ($H_2SO_4$) when emitted into the atmosphere. In the presence of sufficient water vapour, sulphate aerosols ($SO_4$) can be generated, which exhibit the largest cooling effect from aircraft PM emissions [14]. The sulphate aerosol EI is a factor of fuel sulphur content, which varies depending on fuel composition and the specific emissions characteristics of the engine [24]. Due to the low volatility of lubricant oil, the emitted oil vapour from aircraft will add to the condensed mass of VOCs and contribute to vPM concentrations. The VOCs produced either from the oxidized fuel fragments or due to the pyrolysis in the combustion chamber can also act as vPM, which may have sufficiently low vapour pressure to allow condensation in the atmosphere, forming a coating on the surface of the nvPM and impacting cloud formation, precipitation and climate [14]. The particulate matter emitted near the exit nozzle plane of the combustor consists only of nvPM, but vPM are produced through nucleation and condensation downstream. Thus it is difficult to estimate the total amount of vPM produced within the exhaust plume, as the formation of vPM is dependent on the concentration of sulphates and VOCs present in the exhaust, and the distance from the combustor [25].

### 2.2. Aircraft Emissions Modelling

Accurate quantification of aircraft emissions for any given flight requires the calculation of aircraft performance to estimate the total energy consumed, and hence the total fuel burnt throughout the flight duration. With knowledge of fuel flow rates experienced throughout flight, flow rates of aircraft emission species can be deduced based on empirical engine performance datasets and emissions models.

### 2.2.1. Fuel Burn Estimation

Computational modelling of aircraft performance allows the simulation of aircraft trajectories and the quantification of forces experienced throughout flight, enabling the approximation of thrust and fuel flow across all phases of flight. Aircraft performance models that are prevalent in academia and industry, such as the Base of Aircraft DAta (BADA) method [26] and Piano-X [27], approximate aircraft behaviour by coupling a database of aircraft-specific performance datasets to mathematical models to calculate useful flight characteristics, such as fuel flow and fuel burn, at discrete time steps throughout flight. This iterative approach accounts for the time-varying nature of aircraft properties like mass, speed and climb rate, thus leading to a more accurate estimate of performance and fuel burn estimation. Further to this, models such as the Aviation Environment Design Tool (AEDT) [28] exist, which carry out four-dimensional (4-D) physics-based simulations of aircraft trajectories at an exceptionally high spatial and temporal resolution, providing highly accurate predictions of fuel consumption and localised emissions impacts. Such tools do however, come at the expense of high computational and financial cost and often require proprietary data that is not in the public domain. An alternative state-of-the-art open-source performance model that has been made available in recent years is the OpenAP model [29]. This consists of four main components: aircraft and engine properties, kinematic and dynamic performance, and utility libraries; it can describe the characteristics of 27 common aircraft and 400 turbofan engines. The authors present a favourable compar-

ison to established methods, matching BADA's fidelity across many characteristics, and exceeding it in several, deeming it a feasible alternative for researchers who do not have access to data and licensing for proprietary models.

### 2.2.2. Calculating Aircraft Emissions at Reference Conditions

Aircraft performance models provide estimates of aviation fuel burn, but to determine the particular chemical speciation of emissions released due to fuel combustion, aircraft emissions models must be implemented. As mentioned earlier, the primary combustion products, $CO_2$, $H_2O$ and $SO_2$, have a direct relation to the amount of fuel burnt and hence have a constant EI for a given fuel composition. Standard estimates are provided in Table 1. For secondary combustion products however, which vary depending on operational and atmospheric conditions, more complex empirical relationships have been defined. The relationship between EI and fuel flow for secondary products such as $NO_X$, CO and HC have been determined at reference operating conditions, using engine performance datasets, such as the International Civil Aviation Organisation (ICAO) engine emissions databank [30]. This databank was developed for the purpose of engine certification and compliance with landing and take-off (LTO) cycle emissions standards, outlined in ICAO Annex 16 Vol. II [31]. Engine test data have been collected at sea-level static (SLS), International Standard Atmosphere (ISA) conditions, for four reference operating conditions (thrust settings) relevant to the LTO cycle: take-off (100% thrust), climb out (85%), approach (30%) and taxi in/out (7%). For every engine at each of the four LTO modes, a reference fuel flow and corresponding emission index has been derived, allowing emissions to be estimated to a reasonable degree of accuracy for aircraft operating under any of the four modes. An exemplary ICAO EI dataset is provided in Table 2.

**Table 2.** Example ICAO engine emissions data for Rolls Royce Trent 970-84 engine. Data derived from [30].

| Parameter | Take-Off | Climb Out | Approach | Idle |
|---|---|---|---|---|
| Fuel flow [kg/s] | 2.605 | 2.157 | 0.720 | 0.255 |
| EI $NO_x$ [g/kg] | 38.29 | 29.42 | 12.09 | 5.44 |
| EI CO [g/kg] | 0.32 | 0.31 | 1.16 | 13.38 |
| EI HC [g/kg] | 0.02 | 0.12 | 0.08 | 0.04 |

### 2.2.3. Calculating Aircraft Emissions at Non-Reference Conditions

In reality, aircraft spend the majority of flight outside of the LTO vicinity (above 3000 ft), and the operational and atmospheric conditions vary considerably. To enable the accurate analysis of aircraft emissions outside of reference conditions, a number of emissions modelling methods have been developed. Such methods apply the necessary adjustments and interpolations to the LTO-limited engine performance datasets, to generate more realistic estimates of aircraft emissions across the whole flight profile. SAE International Aerospace Information Report 5715 (AIR5715) [32] describes a number of methods for calculating aircraft emissions throughout all modes of operation and compares their relative merits. For the primary combustion species, the Fuel Composition method is presented, which determines $EICO_2$, $EIH_2O$ and $EISO_x$ from the proportions of carbon, hydrogen and sulphur in the fuel. The set of methods concerning the estimation of $EINO_x$, EICO and EIHC include the ICAO reference method, the Boeing fuel flow method 2 (BFFM2), the DLR fuel flow method and the P3T3 method, in order of increasing fidelity. The ICAO reference method serves as the simplest and least accurate approach, computing emissions purely based on the ICAO reference conditions, without applying corrections to account for atmospheric effects at altitude. Therefore, it is only applicable for emissions analysis of aircraft flying in the LTO region. The remaining methods, BFFM2, DLR and P3T3 can be applied at all aircraft operating conditions, including at cruise, as they apply

interpolations to engine performance datasets to determine emissions indices throughout the whole duration of a flight. The BFFM2 method and the DLR method ($NO_x$ only) are the mid-tier models, as they provide reasonable estimates of EIs purely from interpolating the EI reference data and applying corrections for atmospheric effects based on ambient meteorological data, aircraft fuel flow and Mach number. The gold standard for modelling of $NO_x$, CO and HC emissions is the P3T3 method, as it utilises detailed thermodynamic modelling data to determine the precise emission indices at any given point throughout flight. Required data includes the combustor inlet temperature (T3), pressure (P3) and the fuel-air ratio (FAR) at both reference and operational conditions, all of which are difficult to obtain without access to proprietary engine-specific performance data, which limits the accessibility of this model to open-access researchers [33,34].

Figure 1 shows an example of how EIs are interpolated based on the logarithmic relationship between EI and fuel flow, using the BFFM2 method [35]. Furthermore, methods are also presented to account for the remaining key emission species, such as the Derivative Factor method [28] used to approximate the EI values for VOCs such as non-methane HC (NMHC) throughout flight and the First Order Approximation method [23] used to estimate PM emission indices. The choice of method generally depends on the emission species to be observed, the compromise between modelling resources and data availability, and the level of accuracy required.

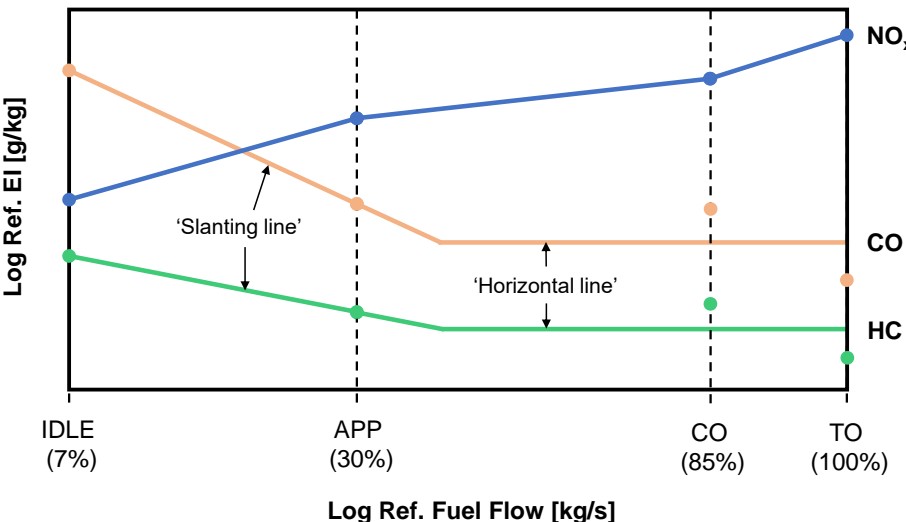

**Figure 1.** Example log-log plots of EI against fuel flow at reference conditions, as prescribed by the BFFM2. Percentage values refer to thrust as a percentage of maximum thrust at SLS conditions for idle (IDLE), approach (APP), climb out (CO) and take-off (TO) conditions.

2.2.4. Emissions Inventories and Integration into Large-Scale Climate Models

The estimation of aircraft emissions for a specific flight involves the simulation of aircraft performance across the entire flight profile so as to estimate fuel flow and engine operating conditions throughout flight. Knowledge of fuel flows and engine performance characteristics permits the estimation of aircraft emissions, based on the coupling of empirical engine certification data and emissions modelling methods, which interpolate the data to determine the emissions at non-reference conditions. This emissions estimation procedure is commonly carried out on a regional and global scale to determine emissions from a whole range of flights, which are then stored in so-called emissions inventories [36]. Aircraft emissions inventories collate data from all flights in the desired range and populate three-dimensional (3-D) latitude-longitude-altitude grid cells (e.g., 1° × 1° × 1000 ft) with total emissions quantities [4].

Aircraft emissions inventories are utilised to model the atmospheric effects of aviation, using large-scale models that capture the chemistry, physics and dynamics of the Earth-

atmosphere system (see Section 5.3 for further detail on climate modelling approaches). Such models allow one to simulate the perturbation to the state of the atmosphere due to an input of emissions and, in turn, provide quantitative indicators that enable modellers to determine the resultant climate impact (e.g., concentrations of key greenhouse gases, aerosol formation and distribution, cloud processes etc.) [37]. One underlying issue with this conventional approach to aviation climate modelling is that the use of gridded emissions data inherently assumes the instantaneous dispersion (ID) of emissions into the latitude-longitude-altitude computational grid cell in which they were released [38]. The dimensions of this grid cell are solely determined by the spatial resolution of the global model, and hence do not serve as an accurate physical representation of emissions dispersion.

In reality, emissions released from aircraft are confined to the aircraft exhaust plume, which inhibits mixing with the surrounding atmosphere for up to a day after emission. Throughout this time, a number of nonlinear chemical and microphysical processes occur, due to the elevated concentrations of emissions species in the plume. These nonlinear plume-scale processes affect the eventual climate response to these emissions, yet most regional and global aviation climate impact studies [39] often neglect the presence of the aircraft exhaust plume and opt for the simplified ID method. The following section explores the dynamical evolution of emissions following their release into ambient air, and discusses modelling approaches present in the literature which can be implemented to represent plume-scale effects in large-scale models. Such modelling is, however, often set aside due to computational issues associated with resolving consistency between the two model resolutions.

### 3. The Dispersion of Aircraft Emissions and the Aircraft Exhaust Plume

Following their expulsion into the free atmosphere throughout flight, aircraft exhaust gases are confined to a plume that undergoes a series of dynamical regimes (**jet**, **vortex**, **dispersion** and **diffusion**), before becoming fully diluted in the surrounding air. The entrainment of emissions within the plume throughout these dynamical regimes leads to initial species concentrations that are several orders of magnitude higher than background levels [40], giving rise to a number of nonlinear chemical and microphysical effects. These plume-scale effects have considerable implications on the eventual chemical composition of the surrounding atmosphere and lead to the formation of aerosols and ice crystals in the aircraft wake. Therefore, inclusion of plume-scale effects is vital for high fidelity modelling of aviation's impact on the climate.

### 3.1. Plume-Scale Dynamical Regimes

In order to accurately account for nonlinear effects experienced in the aircraft exhaust plume, one must first understand the dynamical response of the plume after combustion, to gauge the length and time scales over which aircraft emissions are entrained within it. Figure 2 is a visual representation of the typical dynamical evolution of an aircraft exhaust plume, throughout its lifetime.

Exhaust gas temperatures range from around 1500 K post combustion, 600 K at engine tailpipe, followed by mixing with bypass air cooling the flow to around 300 K [16] as the dispersion process begins. During the first 1–20 s post emission, an axisymmetric jet is formed, which rapidly diffuses into ambient air and cools to ambient temperatures. Over this period, known as the **jet** regime, the airflow passing over the wings is diverted downwards to generate lift, thus creating a vortex sheet at the trailing edge of the aircraft. This vortex sheet rolls up into a pair of counter-rotating vortices which are shed at the wing tips. The evolving vortex pair then merge together and propagate downwards, due to their mutually induced downwash velocity, trapping the exhaust plume within their cores and signalling the beginning of the vortex regime.

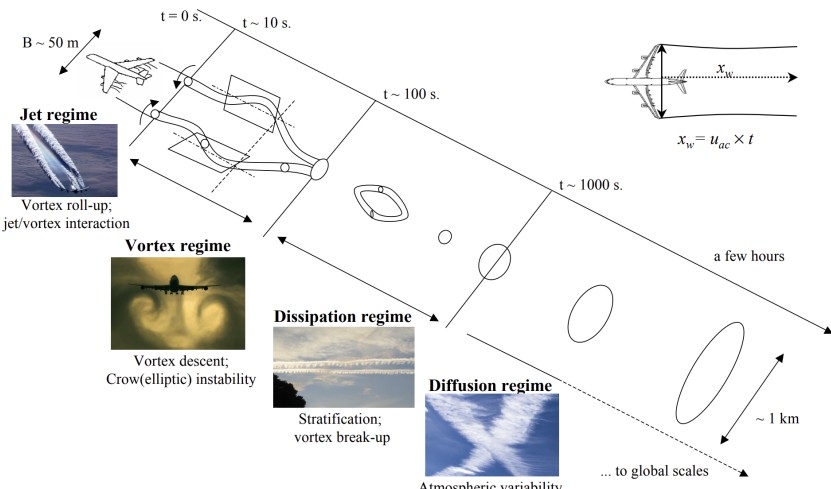

**Figure 2.** Aircraft exhaust plume dynamical regimes. Wake distance behind aircraft $x_w = v_\infty \times t$ where $v_\infty$ is aircraft speed and $t$ is the post emission time. Figure obtained with permission from Roberto Paoli [41].

Throughout the **vortex** regime, which occurs between 20 s and 2 min after combustion, the primary wake containing the vortices and trapped exhaust plume sinks by around 150–200 m, resulting in a slight temperature increase of 1–3 K, due to the adiabatic heating of exhaust constituents in the sinking vortices. Further to this, the organised vortical structure means the wake does not grow significantly during this time, and hence the concentrations of entrained chemical species remain relatively constant. The adiabatic heating of the exhaust does however lead to baroclinicity at the border between each vortex and the ambient air, which detrains some momentum, heat and exhaust constituents from the primary wake, to form a secondary wake. This secondary wake trails upwards as it is warmer than the surrounding ambient air, escaping the influence of the vortex structure and resulting in enhanced mixing with ambient air. Due to its different dynamical evolution, the secondary wake experiences different chemical and microphysical processes to the primary wake [42].

Following this is the **dispersion** regime, in which the aircraft-induced dynamics subside due to the growth of Crow instability [43], which dissipates and disintegrates the primary and secondary wake vortices [41]. The breakdown of the organised vortical structure and the production of turbulent motion leads to a sudden increase in the rate of entrainment between the exhaust plume and the ambient air by a factor of 10, therefore giving rise to a continuous decay of concentration and temperature within the plume. This regime lasts for 2–5 min after combustion, however this varies as the strength of aircraft induced vortices is proportional to the weight and span, and inversely proportional to the speed of the aircraft [42].

Lastly, the plume undergoes its final dynamical event, known as the **diffusion** regime. This regime is characterised by the aircraft-induced dynamics becoming negligible (after about 6 min [44]), followed by the subsequent dominance of atmospheric processes in the spreading of the aircraft exhaust plume and its constituents. Atmospheric turbulence, radiation transport and stratification are examples of natural phenomena that contribute to the diffusion of the plume, with total dilution to ambient concentrations often occurring over timescales of 2–12 h post emission [3]. During this time, the plume may spread up to a few kilometres through atmospheric turbulence and shear in the ambient air, diluting the exhaust species over vast volumes of airspace [45].

Plume-scale climate effects that result from the confinement of emissions to the aircraft exhaust plume during the four dynamical regimes considerably alter the eventual global warming effect of a particular flight, and therefore should be appropriately accounted for in modelling efforts to estimate aviation's climate impact.

### 3.2. Plume-Scale Modelling

To tackle the issue of neglected plume-scale effects in the computational analysis of aviation-induced climate change, a number of plume modelling methods of varying fidelity have been theorised in the literature. Sub-grid resolution plume models simulate the dynamical response of the aircraft plume, so as to capture the nonlinear chemistry and microphysical effects that occur within it. The outputs of plume models can then be parametrised into low-resolution global models, to increase the accuracy of climate impact calculations through better accounting of the emissions dispersion process.

### 3.2.1. Empirical Dilution Model

Plume dynamics control the rate at which aircraft emissions mix and dilute into the surrounding atmosphere, directly affecting the resulting climate impact of aircraft emissions. This is due to the presence of nonlinear effects experienced in the plume prior to becoming homogeneously mixed into ambient air. Quantifying the rate of dilution and modelling the climate effects that occur within aircraft plumes is therefore an essential process in the accurate analysis of aviation's climate impact [45].

In Schumann et al. [46], an empirical dilution model was developed to investigate the mixing rate of plumes throughout their typical lifetimes, based on data collated from over 70 aircraft exhaust plume encounters with research aircraft. The characteristic property observed in this study is the plume dilution ratio, $N$, which is defined as the amount of air mass that the exhaust plume generated from a unit mass of fuel burn mixes with, per unit flight distance within the bulk of the plume. The empirical dilution model is depicted in Figure 3, in which measured dilution ratios from the plume encounters are plotted against plume age. These data were collated through measurements of chemical concentrations across a variety of aircraft types, within the time interval of 0.001–10,000 s. The significance of these findings is that, in spite of the diverse range of chemical species and aircraft types observed, throughout all four dynamical regimes, a relatively consistent logarithmic relationship emerges between dilution ratio and increasing plume age. When interpolating the regression line fitted to the data in Figure 3, the following equation can be obtained, where $N$ represents dilution ratio, $t$ is plume age in seconds, and $t_0$ serves as an arbitrary reference scale

$$N = 7000(t/t_0)^{0.8}, \ t_0 = 1 \text{ s.} \tag{1}$$

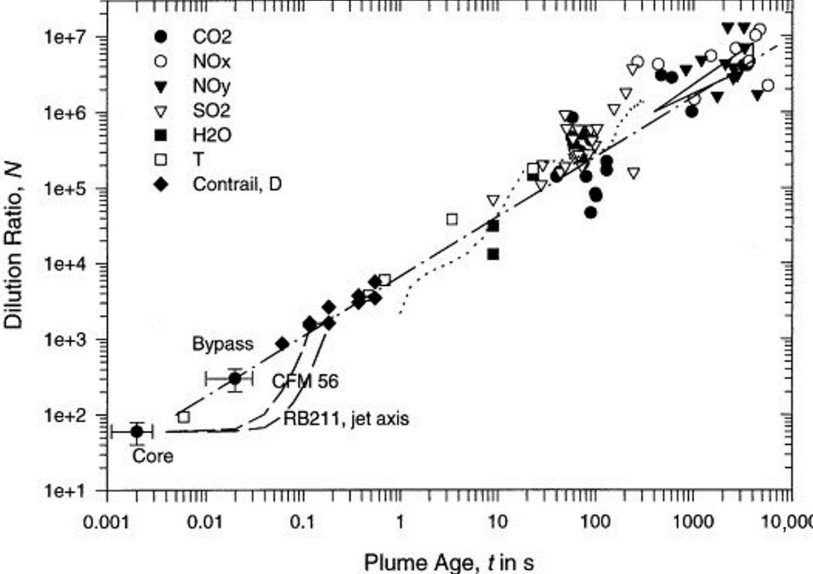

**Figure 3.** Dilution ratio against plume age derived from empirical data. Marker shapes correspond to tracer species as displayed in legend. See original for further marker descriptions. The dash-dotted line shows the interpolation that is represented by Equation (1). Reprinted with permission from Schumann et al. [46]. Copyright 2022, Pergamon.

It is evident from the figure however, that encounters with plumes older than 50–100 s tend to deviate from the line fit considerably, indicating a reduction in accuracy of the logarithmic approximation over time. This is likely due to the transition from the organised vortex structure present in the vortex regime, to the turbulent dispersion regime, where more unpredictable atmospheric processes begin to take place and become the primary influence on the evolution of the plume. Therefore, this empirical model can only be used reliably up to the vortex regime. Beyond this, a Gaussian approximation to the distribution of species concentrations is typically employed, accounting for dispersion effects experienced at cruising altitudes, such as advection, gravitational sedimentation, anisotropic diffusion, wind shear and stable stratification [47]. Two popular plume modelling methods which implement Gaussian approximation and the two-dimensional diffusion equation are the Single Plume model and its discretised counterpart, the Multi-layered Plume model.

### 3.2.2. Single and Multi-Layered Plume Models

The Single Plume (SP) model, first presented in Petry et al. [38], approximates the time-evolving concentration field of an aircraft exhaust plume using a Gaussian distribution [48]. Petry represents diffusion through Equation (2), a differential equation that describes the temporal variation of exhaust concentration $C$ of species $i$ in the plume

$$\frac{\partial C_i}{\partial t} = -sz\frac{\partial}{\partial y}C_i + D_v\frac{\partial^2}{\partial^2 z}C_i + D_h\frac{\partial^2}{\partial^2 y}C_i + 2D_s\frac{\partial^2}{\partial y\partial z}C_i. \tag{2}$$

This two-dimensional diffusion equation is a function of wind shear ($s$), horizontal ($y$) and vertical distance from plume centre ($z$), and the horizontal ($D_h$), vertical ($D_v$) and shear ($D_s$) diffusion coefficients, as calculated based on empirical data recorded under typical atmospheric conditions at aircraft cruising altitudes and assuming a horizontal flight path [49]. The solution to the diffusion equation is a time-varying Gaussian function, with standard deviations $\sigma_h$, $\sigma_v$ and $\sigma_s$ that depend on $s$, $D_h$, $D_v$, $D_s$, time $t$ and the respective initial values $\sigma_{0h,v}{}^2$

$$\sigma_h{}^2(t) = \frac{2}{3}s^2 D_v t^3 + (2D_s + s\sigma_{0v}{}^2)st^2 + 2D_h t + \sigma_{0h}{}^2, \tag{3}$$

$$\sigma_v{}^2(t) = 2D_v t + \sigma_{0v}{}^2, \tag{4}$$

$$\sigma_s{}^2(t) = sD_v t^2 + (2D_s + s\sigma_{0v}{}^2)t. \tag{5}$$

The standard deviations of the Gaussian function can then be used to deduce useful parameters such as plume cross-sectional areas and concentrations. These parameters can then serve as input to atmospheric models, enabling the simulation of the dynamical evolution of the plume, and its entrained emissions throughout its lifetime.

The main drawback of the SP model however, is the assumption of homogeneous concentration distribution throughout the plume at any given time. This homogeneity assumption is sufficiently accurate up to the vortex regime, where plume cross-sectional areas are relatively small, entrainment rates are low, and the mixing ratio is relatively consistent across the plume diameter [50]. However, beyond this, the spike in entrainment rates following the breakdown of vortices causes rapid plume expansion, and the drop in concentration from plume core to outer edges becomes increasingly significant [51,52]. This spatial concentration gradient along the plume cross-section cannot be captured using the SP approach, so an alternative model is proposed, known as the Multi-layered Plume (MP) model, as seen in Figure 4. The MP model builds upon the SP model by discretising the plume cross-section into a number of concentric rings, enabling the inhomogeneous concentration profile to be represented by varying the mean concentration in each ring.

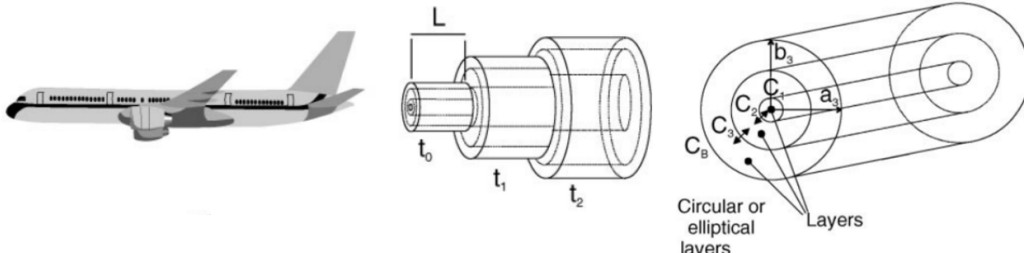

**Figure 4.** MP model visual representation. Plume length L was set to the distance travelled in 1 s (247 m in this scenario) with three out of eight concentric rings shown for each timestep. $C_1 - C_3$ indicate the concentrations in each ring, with arrows representing mixing between the layers. Reprinted with permission from Kraabøl et al. [2]. Copyright 2022, Pergamon.

As Kraabøl et al. [2] states, the Gaussian approximation to emissions dispersion only applies to the dynamical evolution of passive species, and is not suitable for modelling the evolution of chemically active species, due to the nonlinear chemical response in the plume. To counteract this issue, this paper implements an adapted version of the MP model, in which the plume is divided into 8 concentric rings, each with a chemistry module incorporated to estimate the chemical production and loss mechanisms of all species present. Applying this model under assumed turbulent conditions derived from Dürbeck and Gerz [53], graphs of horizontal and vertical plume radius are plotted against plume age, as shown in Figure 5.

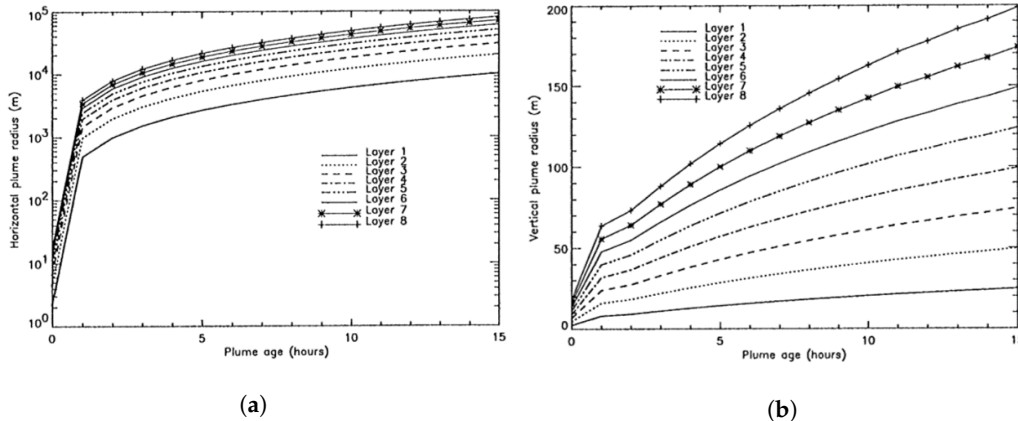

(**a**)                                                 (**b**)

**Figure 5.** Evolution of (**a**) horizontal and (**b**) vertical plume radius over time, for MP model under assumed turbulent conditions. Reprinted with permission from Kraabøl et al. [2]. Copyright 2022, Pergamon.

After just 1 h, it is predicted that plumes can spread between 1 and 10 km horizontally, whilst only reaching around 50–100 m vertically due to atmospheric stratification [49]. As the plume approaches the end of its typical lifetime, between 10 and 15 h, plume cross-sections can reach 100–200 km horizontally and 200–400 m vertically. The vast length scales over which plumes span throughout their lifetime provides ample evidence to suggest that, in high-density airspace, plumes can overlap. The overlapping of plumes can thus lead to spikes in emissions concentrations that exceed that of single aircraft plumes. This in turn, further augmenting nonlinearities in the climate response, thus propagating discrepancies between plume and global model outputs [54].

### 3.2.3. Aircraft Plume Chemistry, Emissions, and Microphysics Model (APCEMM)

Models such as APCEMM from Fritz [55] further increase the accuracy of the SP and MP models, by capturing additional effects that influence plume evolution and hence influence the resultant climate response once the plume is fully diluted. This includes factors

such as plume anisotropy and asymmetry which impact the eventual spatial distribution of the plume, as well as modelling of microphysical processes that strongly affect contrail formation and persistence. The model is similar to the MP model in that a chemistry module is simulated in a number of concentric rings which have differing concentration fields. These concentration fields decrease in magnitude with increasing radial distance from the plume core. The rest of the computation (i.e., mixing and microphysics) is peformed on a high-resolution Cartesian grid [56]. The primary aim of this model development was to bridge the gap between the simplified Gaussian approximation and more comprehensive large eddy simulations, as elaborated on in the following subsection.

3.2.4. Large Eddy Simulations (LESs)

Plume dynamical evolution can be most accurately captured using high-resolution LESs over the entire lifetime of the plume. LESs can model dynamics on a scale of several millions of grid points, for a few seconds to a few minutes of plume age, providing unmatched levels of accuracy at the cost of extremely high computational demand. For this reason, LESs are usually limited to case studies from which the data obtained can be used to derive and calibrate plume parametrisations for use in the lower fidelity methods [57].

In Dürbeck and Gerz [49], LES data are used to calculate effective diffusion coefficients and plume cross-sectional properties for plume modelling purposes. The data obtained from the simulations is said to have agreed with Schumann et al. [45], where the horizontal and vertical plume scales and respective diffusivities are estimated from experimental data captured by a research aircraft measuring NO concentrations. Moreover, LESs have been used to determine plume properties on much shorter timescales in Unterstrasser et al. [44]. In this paper, plume evolution is analysed for the jet and vortex period up to 6 min, whilst aircraft-induced dynamics dominate; concentration profiles and plume cross-sectional areas are determined for a range of atmospheric conditions, varying stratification, turbulence, wind shear and aircraft properties. Similarly in Paoli [58], detailed LES numerical simulations are carried out for the jet and vortex phase, confirming hypotheses surrounding aerosol and microphysical modelling. These studies exemplify the use of LES methods to validate experimental findings, and serve as a means of calibration for plume model parameters that represent real-life plume dilution characteristics.

## 4. Air Traffic and Emissions Distribution

The previous section discussed the forces of flight and the generation and dispersion of aircraft emissions, from the perspective of a single aircraft. However, the state of the atmosphere is inhomogeneous with respect to space and time, meaning that the climate sensitivity to aircraft emissions differs considerably, depending on the time and location of their release. Furthermore, in airspace regions where traffic density is high, aircraft fly in close proximity, meaning their exhaust plumes may intersect and merge. This superposition of plumes allows concentrations of emissions contained within them to accumulate, accentuating nonlinear plume-scale effects that must also be considered in aviation climate analysis. Due to the spatio-temporal sensitivity of non-$CO_2$ climate impact, and the potential for plume interactions between aircraft, it is therefore useful to explore real world air traffic distribution with respect to both time and location. This section delves into the influence of air traffic management and aviation passenger demand on both the global and local distribution of air traffic, and hence associated emissions.

*4.1. Air Traffic Management*

Air traffic management (ATM) is the system of services responsible for overseeing the network-wide implementation of safe, orderly and efficient air traffic flows, providing assistance to aircraft in transit from departure to destination aerodrome. It is the role of the air traffic control (ATC) team to manage and monitor air traffic in their respective airspace in real time, ensuring that optimum safety, order and efficiency of aircraft operations are maintained at all times [59].

### 4.1.1. Air Traffic Safety

The inherent risk involved in the transportation of vast numbers of passengers at near transonic speeds through the upper atmosphere means that aviation safety is of paramount importance. A safe aircraft operation takes the path of least danger, primarily influenced by the need to avoid unfavourable atmospheric conditions and to prevent conflicts with other aircraft [60,61]. In-flight atmospheric conditions susceptible to icing, turbulence or the presence of hazardous convective weather can all be classified as unfavourable for aircraft, with the latter presenting the greatest constraint on aircraft routing [62,63]. The increased risk resulting from flight through weather-affected regions means that aircraft must re-route, leading to restrictions on available airspace and deviations from the optimal flight profile. Consequentially, this results in greater flight times, fuel burn and delays for aircraft subject to such conditions [64].

The safety risks associated with aircraft-to-aircraft collisions necessitate air traffic controllers to impose safe separation standards between aircraft in the lateral, longitudinal and vertical direction, as specified in [65]. The stated minimum separation distances between aircraft are 5 nautical miles (NM) laterally, 20 NM longitudinally and 1000 ft in the vertical direction under the most lenient scenarios. A breach of separation laws in more than one direction is known as a conflict and must be resolved as quickly as possible. The enforcement of separation minima therefore introduces a theoretical upper limit on the number of aircraft that occupy a fixed volume of airspace at any one time, otherwise known as airspace density.

### 4.1.2. Air Traffic Order

To keep air traffic flows organised within controlled airspace, aircraft are ordered to follow the traditional fixed-route air traffic network, constructed from four key airspace elements that facilitate the air traffic management process [59]:

- **airports/aerodromes**—an area of land or water intended to be used for the arrival, departure and surface movement of aircraft;
- **waypoints**—a specified geographical location used to define the flight path of an aircraft, representing either a navigational aid (navaid) or a reference coordinate that the aircraft must fly by or fly over;
- **airways**—a controlled portion of airspace established in the form of a corridor (usually 8–10 NM wide) between two waypoints;
- **sectors**—a region of airspace managed by a single ATC team, stratified into various levels to accommodate a wide variety of traffic.

In [66], the notion of optimising air traffic flows for a given demand and capacity is explored, in which air traffic flows are represented using these four key elements. **Airports** represent the sources and sinks of the flow, **airways** are the arcs along which the flow travels, **waypoints** are the network nodes at which airways intersect, merge or diverge, and **sectors** are a collection of waypoints and contiguous segments of airways. The fixed-route network restricts airspace availability even further, due to the discretisation of flight levels and the requirement to pass specified waypoints [67,68]. This can lead to particularly high frequencies of aircraft passing through high-density airspace, potentially leading to congestion along busy airways and waypoints where airways intersect, resulting in inhomogeneities in the distribution of air traffic and potential congestion along high-density airways.

### 4.1.3. Air Traffic Efficiency

The third and final component of effective air traffic management is the optimisation of flight trajectories, subject to the prioritisation of safety and the compliance with the fixed-route airspace structure. Flight trajectory optimisation is an essential step in ensuring maximum airspace utilisation and efficiency, so that revenue is maximised and demand levels are sufficiently met. Trajectory optimisation is a multi-faceted problem, requiring

consideration of nonlinear aircraft performance, wind and weather forecasts, payload, departure fuel load, reserve fuel load and ATM constraints that restrict aircraft operations and routing [69]. This requires an exhaustive assessment to be carried out at the flight planning stage, to test all possible combinations of route, payload, fuel load and operating approach, involving tens to hundreds of thousands of calculations per flight. The most optimal scenarios are then ranked in order of optimality, with the final route selected based on operator preference and/or the occurrence of unexpected circumstances, such as sudden adverse weather conditions or aircraft conflicts [70].

In an ideal airspace situation where the atmosphere is calm and constant; aircraft are not constrained to a fixed route; and there is no risk of conflict with other aircraft, the least-time and least-energy aircraft operation would be to fly the great-circle arc between departure and destination. The vertical profile of the aircraft would consist of a continuous climb out to the most efficient cruise altitude, then to cruise at constant speed, with the ability to cruise-climb continuously as the aircraft burns fuel and loses mass. In reality, the true optimal route can deviate considerably from the great-circle arc, instead taking the path which minimises the risk of bad weather encounters and collisions, abides by the fixed-route airspace structure, whilst also flying a route which is optimised with respect to wind and temperature. The magnitude and direction of wind and the localised variation in temperature experienced by the aircraft throughout flight, can have a drastic impact on route optimality, with tailwinds and colder temperature regions being favourable [71]. Ng et al. [72] found for a wide range of wind-optimal flight scenarios that domestic flights saved up to 3%, and international flights saving up to 10% on both fuel burn and travel time, despite flying longer routes. Furthermore, the vertical flight profile of the aircraft must adhere to flight level allocations, meaning that step climbs must be performed as fuel is burnt, further condensing air traffic and its corresponding emissions into narrow bands of altitude.

### 4.1.4. Airspace Capacity

The effective management of air traffic relies on the human cognition of air traffic controllers to make difficult decisions and carry out complex tasks in a time-critical dynamic environment. This includes ensuring safety through avoidance of poor atmospheric conditions and conflicts with other aircraft, maintaining order by flying along the fixed-route airspace network, and optimising air traffic flows with respect to wind and weather.

As density and complexity levels of air traffic increase, so does the mental workload of the air traffic controller, up until a threshold level is reached where the controller can no longer safely handle the situation. The maximum number of aircraft permitted by the ATC team in charge of a particular airspace volume is known as the airspace capacity, and is driven by the airspace situation, state of equipment being used, and the controller's own mental state [73]. Airspace capacity is limited more by controller workload than it is separation laws, meaning human cognition is the true limiting factor on the number of aircraft that can occupy a particular airspace volume at a particular time [67,74]. Therefore, models of controller workload are often used to estimate airspace capacity, in which ATC tasks are modelled to determine a safe upper limit on workload. In Welch et al. [74], a macroscopic workload model is proposed which generalises ATC tasks into four distinct categories: background, transition, recurring and conflict tasks. This provides an objective basis for estimating capacity and enables the formulation of an analytical relationship between airspace capacity and sector volume, as seen in Figure 6.

The capacity estimation model from Figure 6 predicts that, for a 10,000 NM$^3$ rectangular sector of dimensions 156 NM (length, ratio 4:1) × 39 NM (width) × 10,000 ft (height), a maximum of 16 aircraft may be present at any one time. In a purely hypothetical situation where separation laws dictate capacity and all aircraft are travelling in one direction lengthways, the sector could support a maximum of 490 aircraft, assuming a separation of 5 NM laterally, 20 NM longitudinally and 1000 ft vertically.

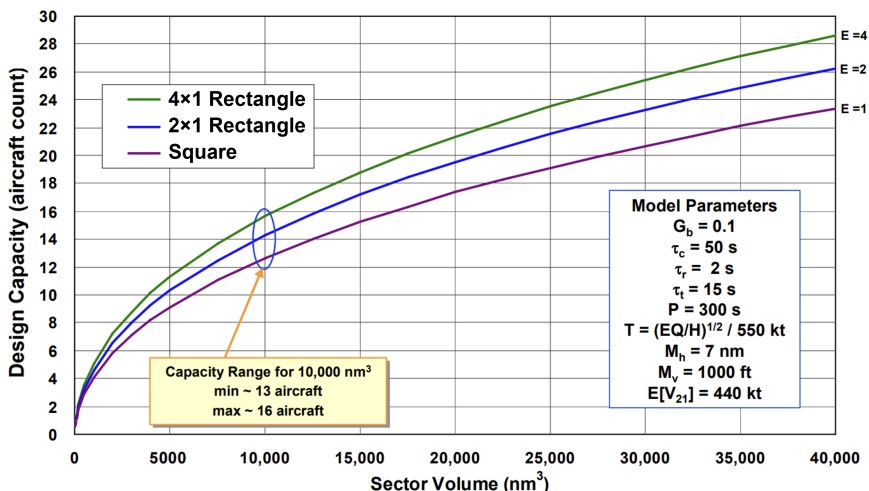

**Figure 6.** Aircraft capacity estimation against sector volume for a range of airspace scenarios. E refers to the length-to-width ratio of the sector, $G_b$, $\tau_C$, $\tau_r$ and $\tau_r$ are all empirical parameters related to controller workload, P is the mean task recurrence period per aircraft, T is the sector transit time, $M_h$ and $M_v$ are the designated horizontal and vertical separation minima between aircraft within the sector and $E[V_{21}]$ is the typical aircraft closing speed [75]. Figure obtained from Welch et al. [74].

The comparison between expected and hypothetical maximum capacity emphasizes the limitations to air traffic density imposed by human factors. This highlights the need for airspace modernisation to increase automation, integration and collaboration in the ATM system, enabling the further increase in capacity levels towards minimum separation capacity [76].

### 4.2. Global Air Traffic and Emissions Distribution

The nature of air traffic and emissions distribution was investigated in Olsen et al. [77] where a range of global aircraft emissions datasets are compared (NASA-Boeing 1992, NASA-Boeing 1999, QUANTIFY 2000, Aero2k 2002, AEDT 2006 and aviation fuel usage estimates from the International Energy Agency) to show distribution patterns in the latitudinal, longitudinal and vertical sense. Further to this, temporal variations with respect to both the diurnal (time of day) and seasonal (time of year) cycles are explored.

The climate sensitivity of the atmosphere is highly variable depending on the exact latitude, longitude and altitude combination, because of the spatially varying chemical and meteorological state of the atmosphere (e.g., [78–80]). Accurate accounting of spatial distribution patterns of air traffic is therefore very important in the estimation of aviation climate impact. Figure 7 shows the spatial distribution of fuel burn across the range of datasets in the longitudinal, latitudinal and vertical directions. The longitudinal distribution shows three emissions peaks around the densely-populated regions of the US, Europe and East Asia, with the largest situated above the North American land mass. It is evident in the latitudinal distribution, that the Northern Hemisphere dominates, with a strong peak in the northern mid-latitudes that appears due to high volumes of air traffic above the US and Europe, as well as along the connecting region of airspace, the North Atlantic flight corridor (NAFC). Contrarily, there are almost no emissions present in southern latitudes below 40° S, with the region between 40° S and the equator constituting only a small percentage. The altitudinal distribution on the other hand, experiences emissions peaks around both the low altitude LTO area and the high-altitude cruising regions between 9 and 13 km, with relatively low emission intensities at mid-altitudes. Furthermore, the peak around cruising altitude is discretised into peaks every other flight level, due to the vertical separation constraints and the allocation of aircraft to specific flight levels, thus owing to further increases in emissions intensities at these altitudes.

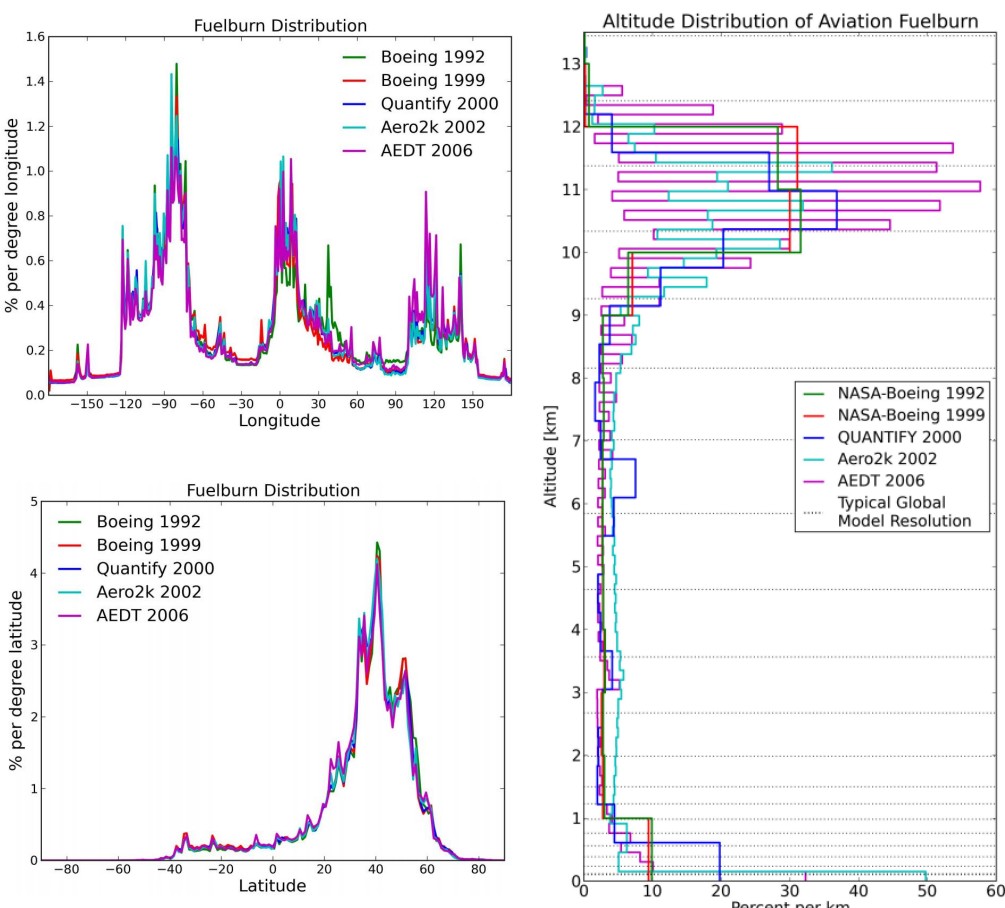

**Figure 7.** Spatial (latitude, longitude and altitude) distribution of global aviation fuel burn from a range of aircraft emissions datasets. Figure obtained from Olsen et al. [77] licensed under CC BY 3.0.

The presence of diurnal and seasonal variations in key chemical and meteorogical parameters throughout the atmosphere has been widely investigated in the literature (e.g., [81]). The diurnal and seasonal variation in aviation fuel burn from Olsen et al. [77] is displayed in Figure 8. The temporal fluctuations in both the state of the atmosphere and the distribution of fuel burn and emissions allude to the fact that the climate sensitivity to aircraft emissions is always changing, and therefore these parameters must be under constant observation to ensure accurate determination of global climate effects from aviation.

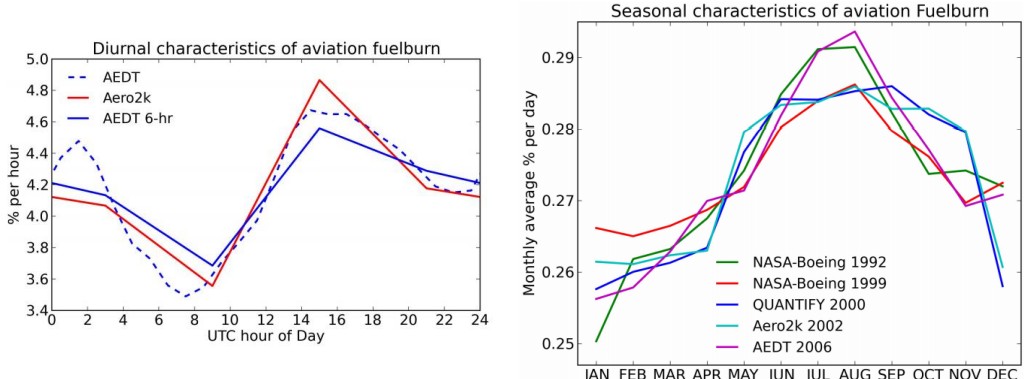

**Figure 8.** Temporal (diurnal and seasonal) distribution of global aviation fuel burn expressed as percentage from a range of aircraft emissions datasets. Figure obtained from Olsen et al. [77] licensed under CC BY 3.0.

The diurnal cycle of global aviation fuel burn, as seen on the left hand plot of Figure 8 displays a peak at around 15:00 UTC, which decreases through the night until around 09:00 UTC where total fuel burn begins to increase again. With regards to seasonal variation, there is significant variance between emissions datasets, however in general, all display a wintertime minimum between December and January, and a summertime maximum between June and September.

*4.3. Local Air Traffic and Emissions Distribution*

Due to the fixed-route nature of airspace, aircraft tend to fly along common airways or flight corridors, and pass common waypoints along their journey, leading to exceptionally high flux densities of aircraft through these regions at peak times. This has implications on the nonlinear chemical and physical effects occurring at the plume scale, due to the intersection of aircraft plumes and the elevated exhaust gas concentrations entrained within them. A prime example of a high-density airspace region is the NAFC, made up of a series of tracks that aircraft traversing the North Atlantic must follow, updated daily to allow for convective weather avoidance, tracking of the North Atlantic Jet Stream and favourable tailwinds to maximise efficiency [82]. The annually-averaged number of aircraft traversing the NAFC per day has increased from 800 in 1997 [83] to around 2500 in recent years [84], owing to the tripling of passenger demand since then [85]. With North Atlantic air traffic being confined to a limited number of tracks (usually three or four), it can be assumed that aircraft separation distances and airspace capacities are pushed to their limit on a regular basis along this and other popular flight corridors around the world.

Previous experimental work on air traffic emissions in the NAFC was carried out in the late 1990's, through campaigns such as the Pollution from Aircraft Emissions in the North Atlantic Flight Corridor (POLINAT) and Subsonic Assessment Ozone and Nitrogen Oxide Experiment (SONEX) [86]. At least 20 follow up papers were published following these campaigns, in which POLINAT/SONEX data are utilised to provide insight on a number of major scientific issues [87]. A noteworthy publication with regards to localised emissions impacts is Schlager et al. [54], which carries out an in-situ investigation of air traffic emissions signatures (nitrogen oxides ($NO_x$), sulphur dioxide ($SO_2$) and cloud condensation nuclei (CCN)) in the NAFC using experimental data from a POLINAT research flight. The research aircraft flew perpendicular to the major eastbound corridor tracks and took measurements of various chemical concentration fields and meteorological parameters throughout. The results show that the superposition of aircraft exhaust plumes led to peak concentrations of $NO_x$, $SO_2$ and CCN above background levels by factors of 30, 5 and 3, respectively. This is because plume dispersion timescales greatly exceed the daily frequency with which aircraft emissions are input into the flight corridor, resulting in an inhomogeneous concentration field with narrow and sharp peaks over a relatively low and smooth background level [86].

In the observation of plume-scale climate effects in high-density airspace regions, aircraft separation minima determine the minimum possible distance between aircraft and hence the maximum possible overlap between aircraft exhaust plumes. The degree of plume overlap influences the magnitude of emissions saturation, further accentuating nonlinear plume-scale climate processes that occur, as elaborated on in Section 6.4. Using plume dimension estimates from Kraabol et al. [2], maximum plume overlap scenarios can be inferred. Figure 9 displays the lateral, longitudinal and vertical maximum overlap scenarios for aircraft cruising at 550 kt. The longitudinal separation scenario proves to be the most effective formation for superimposing aircraft plumes, with the intersection time simply equalling the time taken to travel the separation distance between the two aircraft.

For lateral and vertical superposition, the intersection time is much longer, as it is determined by the plume expansion rate in that particular direction, up until the midpoint between the two aircraft is reached. For lateral plume overlap to occur, the plumes must expand horizontally to a radius of 2.5 NM (15,190 ft), whereas for vertical, the distance is a mere 500 ft. Despite the drastic reduction in distance, the vertical plume overlap takes about

an order of magnitude longer than lateral, because vertical plume expansion is substantially suppressed in comparison due to stable atmospheric stratification counteracting vertical motion [45]. In reality, air traffic flows are much more complex and the controlled and orderly formations shown in Figure 9 are unlikely to occur naturally. However, the premise still holds that plume overlap occurs most frequently in congested flight corridors such as the NAFC, when aircraft travel along similar tracks and the vertical displacement between aircraft is minimal.

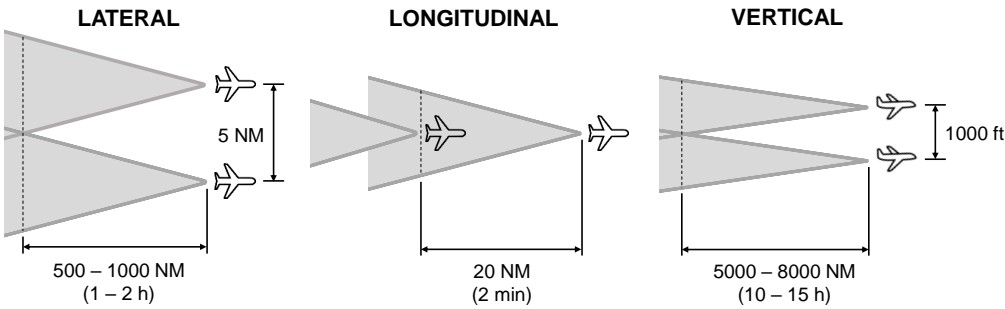

**Figure 9.** Lateral, longitudinal and vertical maximum plume overlap scenarios, based on nominal separation minima [65].

The importance of safety, order and efficiency in the air traffic management process and the characteristics of global and local air traffic flows have been discussed. As the following section will explore, the atmospheric response to aviation emissions is highly sensitive to time and location, because the instantaneous state of the atmosphere (i.e., the chemical composition and meteorological situation at that particular time and position) determines the production, loss and radiative response of key chemical species that induce climatic effects [37]. With insight into how air traffic and emissions are dispersed locally and distributed globally, the contribution of aircraft emissions to climate change can be more rigorously evaluated.

## 5. The Global Climate Impact of Aviation

Aircraft emissions drive climate change by perturbing the flux of inbound shortwave (SW) solar radiation and outbound longwave (LW) terrestrial radiation emitted from the Earth's surface, through absorption and scattering processes that give rise to warming or cooling of the atmosphere [88]. Climate metrics such as radiative forcing can be used to measure the climate contribution of each individual emission species, enabling the determination of the net global warming effect from aviation. This section explores the potential global impact of aviation deduced from measurement and modelling of atmospheric processes.

### 5.1. Aircraft Emissions in the Upper Troposphere and Lower Stratosphere

As Figure 7 suggests, the vast majority (~60%) of aviation fuel burn and hence aviation emissions, occur at cruise altitudes, between 9 and 13 km vertically. The region of the atmosphere encompassing this volume of airspace is known as the Upper Troposphere and Lower Stratosphere (UTLS), with bounds of ±5 km above and below the conventional tropopause [89]. Around 20–40% of total aircraft emissions are released in the LS [14,90] and the rest are released in the troposphere, extending from the surface at take-off to the UT at aircraft cruise altitudes. The greenhouse effect due to the release of chemically-active substances in the UTLS is considerably greater than that of emissions at the surface. This is because the climate in the UTLS is more sensitive due to increased residence times of pollutants, lower background concentrations (meaning emissions have a greater influence on the atmospheric chemistry), lower temperatures which drive reversible reactions in an unfavourable direction, and finally, a higher radiative efficiency which gives rise to more efficient photochemistry [91]. Johnson et al. [92] further validates this claim, with model

results concluding that $NO_x$ emissions constitute a 30 times greater climate impact in the UT compared to equivalent surface emissions, due to the absence of direct deposition and slower conversion to stable reservoir species at aircraft cruising altitudes.

Despite such a large proportion of air traffic emissions being released into the LS region, it is thought that the perturbation to the chemical and radiative state of the stratosphere is negligible. This is because the vast majority of species emitted into the stratosphere are transported downwards into tropospheric regions, where they interact with the atmosphere there [93–95]. Henceforth, this review will primarily focus on the tropospheric response to aircraft emissions, except for water vapour, where the stratospheric climate response becomes particularly noteworthy.

*5.2. Radiative Forcing of Aircraft Emissions*

High altitude emissions from aviation impact the climate through a variety of climate forcing pathways. Some greenhouse gases such as $CO_2$ and $H_2O$ are emitted directly, whereas others are produced indirectly through chemical processing of aircraft emissions, such as the reaction of $NO_x$ with atmospheric trace species to catalyse ozone production and methane destruction. Water vapour and PM emissions are responsible for the formation of high ice clouds known as condensation trails (contrails), which often trap outbound LW radiation within the atmosphere more efficiently than they reflect inbound SW radiation. PM emissions also have the potential to induce a climate perturbation through direct radiative processes due to aerosols, which may warm or cool the climate depending on the particle's optical and microphysical properties [93]. Diversity in the climate forcing pathways for each emission species means that the only reliable method of determining the severity of each climate contribution is to model the atmospheric response and to deduce the resulting impact on radiative fluxes present in the UTLS [16].

The most common climate metric used to compare the magnitudes of climate impact from a range of emission species is radiative forcing (RF). RF is defined as the perturbation to the net energy balance of the Earth-atmosphere system due to natural or anthropogenic factors of climate change, measured in watts per square metre [$Wm^{-2}$] [96]. A positive RF means that the climate forcing mechanism is inducing a warming effect and vice versa. Lee et al. [1] presents an updated analysis of the global effective radiative forcing (ERF) contributions for aviation-induced climate change. ERF is a newly proposed climate modelling framework that builds upon the RF concept by removing rapid atmospheric adjustments that bear no relation to the long-term surface temperature response that occurs over decadal timescales [88]. ERF serves as a more suitable equivalency metric to compare the global warming response of heterogeneously distributed short-lived climate forcers and uniformly distributed long-lived climate forcers. Figure 10 displays the ERF and RF contributions for each of the key aviation-induced climate forcers.

5.2.1. $CO_2$

The climatic effects induced by aviation $CO_2$ emissions are direct and well understood, with the thermal absorption of outbound LW radiation leading to warming through the planetary greenhouse effect [97]. Carbon dioxide from aircraft constitutes the second largest ERF term in figure 10, at 34.3 $mWm^{-2}$, and the thermodynamic and photochemical stability of $CO_2$ means it has a relatively long atmospheric lifetime, on the order of 100 to 1000 years [98]. Therefore, carbon emissions from aircraft tend to become distributed over global spatial scales and accumulate in the atmosphere, leading to increased atmospheric concentrations of the substance over time. The ubiquitous and intuitive nature of $CO_2$-related warming deems it a suitable benchmark to compare warming from non-$CO_2$ climate forcers against. The assumption of instantaneous dilution in global models is sufficient for modelling the climate impact due to carbon dioxide, as it exists over vast spatial and temporal scales, meaning the climatic effects occurring on plume time scales are negligible compared with the impact induced over its entire lifetime [4].

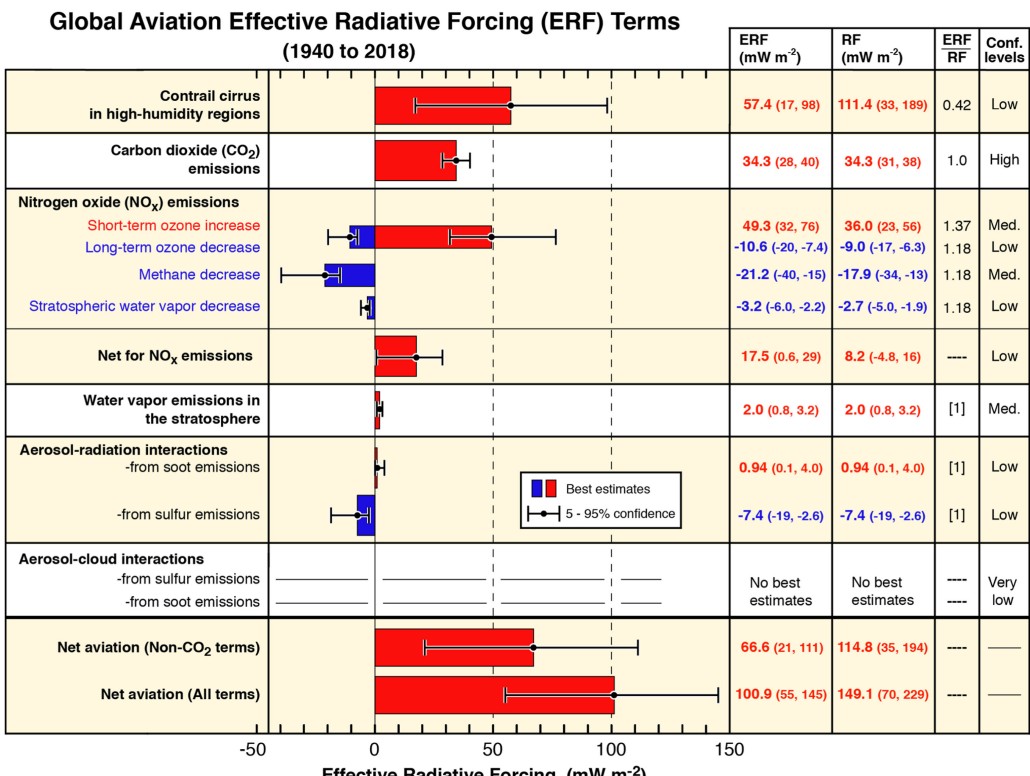

**Figure 10.** RF contributions from global aviation between 2000 and 2018. Error bars represent 5–95% confidence interval, with corresponding error bounds shown in parentheses. Reprinted with permission from Lee et al. [1]. Copyright 2022, Pergamon.

### 5.2.2. Contrail Cirrus

When emissions of water vapour and PM are released into the aircraft exhaust plume in the cold and moist conditions of the UTLS, the water vapour condenses around the particulates due to the high relative humidity (RH), then freezes due to the low external temperatures, permitting the formation of ice crystals in the aircraft wake. This accumulation of ice crystals gives rise to the generation of a condensation trail, also known as a contrail. The evolution of contrails, i.e., how they grow, disperse and persist with time, is largely determined by the ambient conditions of the surrounding atmosphere, with lower temperatures and higher humidities generally leading to more persistent and damaging contrails [99]. The formation and persistence of contrails can be predicted purely from the thermodynamic assumption that, as the hot and moist exhaust mixes with the colder and drier ambient air, a contrail will form if the plume exceeds water-saturation at any point and the temperature is low enough for ice nucleation to occur. A contrail will persist in the atmosphere if it mixes with air that is supersaturated with respect to ice, i.e., an ice-supersaturated region (ISSR), growing with time due to deposition of surrounding water vapour onto existing ice crystals [99,100]. Persistent contrails can sometimes transition to contrail cirrus, either building onto existing cirrus clouds or forming new ones. Contrail cirrus can spread over vast swathes of the atmosphere [101], inducing significant radiative effects that exacerbate their climate impact. ISSRs occur most frequently at aircraft cruising altitudes and often span hundreds of kilometres horizontally, however they only reach depths of 100 to 1000 m [102,103]. Their shallow nature means that aircraft can avoid them by changing flight level by ±2000 ft to minimise persistent contrail generation, for a minor additional fuel penalty of ~1–2% [104].

The optical characteristics of contrail ice crystals often exhibit significant levels of opacity, tending to absorb outbound terrestrial radiation more efficiently than they reflect inbound solar radiation. Contrails and contrail cirrus induce a globally averaged ERF of 57.4 mWm$^{-2}$. This means that in spite of their relatively short lifetime, contrail warming

contributes more to climate change than the accumulation of aviation carbon emissions since the dawn of civil aviation in the 1940s [105]. Contrail evolution, i.e., how they grow, disperse and persist with time, is largely determined by the ambient conditions of the surrounding atmosphere, with lower temperatures and higher humidities generally leading to more persistent and damaging contrails [99].

The formation and persistence of contrails can be predicted purely from the thermodynamic assumption that, as the hot and moist exhaust mixes with colder and drier ambient air, a contrail will form, assuming the plume exceeds water-saturation at any point and air temperature is low enough for ice nucleation to occur.

Contrails exhibit radiative forcing through the obstruction of both SW and LW radiative fluxes, with areal coverage and optical depth (opacity) being the key drivers of contrail climate impact [106]. A contrail's emissivity (its ability to absorb and re-emit infrared LW radiation back towards Earth) and reflectance (its ability to reflect inbound SW radiation back out to space due to scattering at visible wavelengths) is a function of contrail optical depth and ice particle microphysics. See Figure 11 for a visual depiction of contrail forcing caused by emissivity and reflectance. Contrails and other high ice clouds often warm the climate, as their thin optical depth means partial transparency to solar radiation, whilst their high ice density traps infrared radiation within the atmosphere effectively [100]. Contrail RF also displays a distinct diurnal trend; at night, contrails always induce a warming effect, as there is no SW scattering to counteract the LW absorption. During the day, contrails display a reduced net warming effect and perhaps even a net cooling effect, depending on the amount of solar radiation that is redirected back out to space [107].

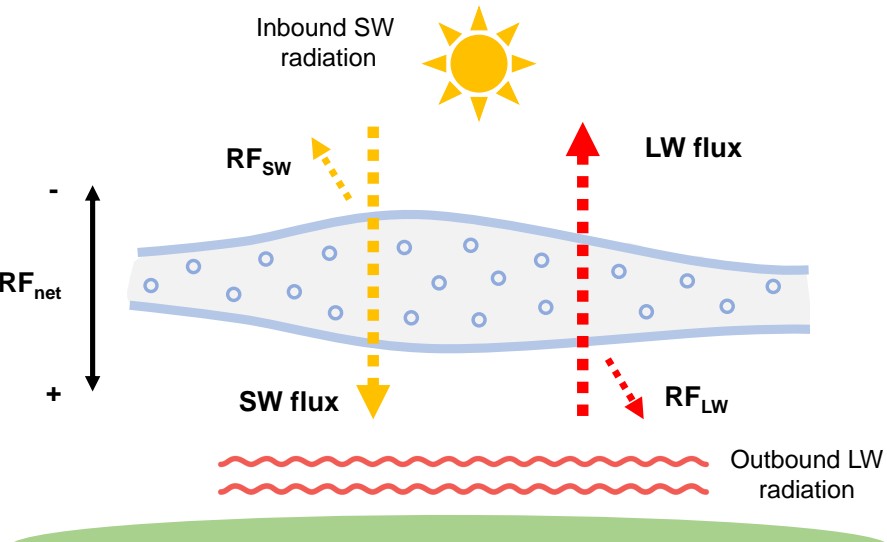

**Figure 11.** Visual representation of contrail radiative forcing. The net RF constitutes the sum of the LW (terrestrial) and SW (solar) RF components.

The so-called Schmidt-Appleman criterion can be used to predict formation and persistence of contrails [108–110], however accurate quantification of contrail RF requires sufficiently detailed microphysical modelling, to deduce key information on the underlying formation mechanisms and physical and optical properties the contrail exhibits throughout its evolution [111]. See Section 6.3 for elaboration on the microphysical processes that determine a contrail's radiative characteristics, and how these processes can be parametrised for contrail modelling purposes.

### 5.2.3. Net-NO$_x$

Oxides of nitrogen (NO and NO$_2$) released from aircraft are not radiatively active and therefore do not induce an immediate climate impact at the point of emission. Their

chemical instability does however mean that they exhibit a number of indirect RF effects, due to the chemical reactions that occur following dilution into the ambient atmosphere. The chemical interactions between $NO_x$ and trace species in the background atmosphere at aircraft cruising altitudes are highly non-linear and thus, the net-$NO_x$ RF contribution is dependent on the instantaneous atmospheric state (time of day and year, latitude, background chemical composition, meteorological situation) [50,112].

Emissions of $NO_x$ in the troposphere initially lead to the short term local increase in ozone production efficiency ($O_3$) on the time scale of weeks to months. In addition, elevated $NO_x$ and $O_3$ levels lead to increased hydroxyl radical (OH) production, which in turn, leads to the long term global destruction of ambient methane ($CH_4$) over the time scale of decades [113–115]. The short term increase in $O_3$ generates a strong positive (warming) ERF of 49.3 mWm$^{-2}$, whereas the long term $CH_4$ depletion causes a lesser negative ERF of $-21.2$ mWm$^{-2}$ in comparison, however there are a number of secondary negative (cooling) radiative forcing contributions arising from methane depletion that must also be accounted for: the reduction in stratospheric water vapour (15% of the $CH_4$ RF magnitude) and a decrease in long-term background ozone in the troposphere (45% of the $CH_4$ RF magnitude), resulting from reduced background $CH_4$ concentrations [116,117]. Despite the long-term negative cooling effects, the short term warming from $O_3$ dominates, leading to a largely positive net ERF of 17.5 mWm$^{-2}$ overall, as seen from Figure 10. Section 6.1 explores the nonlinear $NO_x$-$O_3$ relationship further through explanation of the gas-phase photochemical processes that begin in the aircraft plume and are eventually distributed to regional and global scales.

### 5.2.4. Water Vapour

The direct radiative effects induced by aviation water vapour emissions in the troposphere are insignificant, because the influence on background concentrations is negligible when compared with the natural fluxes of the Earth's hydrological cycle [14]. Any tropospheric water vapour emissions tend to get lost through deposition, due to high humidity and precipitation in this region, leading to a lifetime of around 9 days. On the contrary, water vapour that is emitted into the stratosphere (without getting transported downwards into the UT) can have a prominent influence on the surrounding atmosphere, due to extreme dryness at these altitudes [118]. This is because increases in stratospheric water vapour (SWV) concentrations impact the climate directly through the greenhouse effect, as well as through influences on the gas-phase and aerosol chemical composition. This subsequently leads to depletion of ozone and alters the formation and growth of polar stratospheric clouds [119]. Both the direct greenhouse effect and the indirect impact on ozone and polar stratospheric clouds, caused by increased SWV, leads to an overall ERF of 2.0 mWm$^{-2}$.

As proposed in Lee et al. (2010) [93], the relatively small climate perturbation due to aviation-induced SWV has the potential to increase drastically as future flight concepts begin to take shape. This includes the environmental implications associated with the potential replacement of the current subsonic aircraft fleet with a new supersonic high-speed civil transport fleet, that primarily operates in the stratosphere (e.g., [40,120–122]). Furthermore, the prospect of transitioning the entire subsonic kerosene-based commercial fleet to a hypothetical fleet of cryoplanes (hydrogen-powered aircraft with zero carbon emissions) would increase aviation $H_2O$ emissions by a factor of ~2.5 [123]. In the high-humidity conditions of the UT, this brings about the potential for large increases in contrail production, whereas in the LS, this would induce significant perturbations to SWV concentrations.

### 5.2.5. Aerosol Effects

Aviation aerosol particles that are either emitted directly post combustion, or those which form downstream in the aircraft wake, can perturb the energy balance of the atmosphere directly, as well as indirectly through the formation of contrail ice particles and heterogeneous chemical processing. The direct aerosol effect is primarily produced by the key non-volatile and volatile PM emissions: soot and sulphate aerosols.

Soot exhibits a direct radiative forcing as it has the strongest absorption of light at visible wavelengths per unit mass, more than any other abundant substance in the atmosphere. As a result, it contributes to global heating through the absorption of both inbound solar radiation and light which has rebounded off reflective surfaces such as snow and ice [124]. The resultant heating of the atmosphere and reduction of sunlight can affect the hydrological cycle and large-scale circulation patterns, having potentially larger implications on the climate than previously thought. Despite its ability to strongly absorb sunlight, aircraft soot is responsible for only a few percent of total atmospheric black carbon, meaning the aerosol-radiation interaction brought about by aviation soot only constitutes a minor ERF of 0.94 mWm$^{-2}$.

Sulphur dioxide ($SO_2$) is formed when sulphur, which is present in hydrocarbon jet fuels, oxidises during the combustion process [125]. $SO_2$ that is emitted from the aircraft exhaust can be oxidised to form gaseous sulphuric acid ($H_2SO_4$), which may condense on existing particles or contribute to new particle formation in low-condensation environments, resulting in sulphate aerosol formation. Sulphate aerosol is mainly composed of sulphuric acid and corresponding salts such as ammonium sulphate. The optical properties of sulphate mean that it tends to scatter inbound sunlight, thus leading to a net negative (cooling) ERF of 7.4 mWm$^{-2}$, that sways the net direct climate impact of aviation aerosols towards cooling.

It is relatively well established that aviation-induced aerosols have a direct impact through radiative interactions, as discussed in this section, as well as indirectly through activation of water vapour on PM emissions through contrail formation. There are however, potentially large indirect consequences of aerosol particles interacting with cloud droplets and ambient ice particles nucleation on the aerosol surface. These effects are left without ERF estimates in Figure 10, as there is great uncertainty around the accuracy of cloud process modelling and the ability to distinguish aircraft-induced clouds and natural clouds [126].

Figure 12 is a visual depiction of the typical climate forcing contributions due to aircraft emissions, as described in Lee et al. [1].

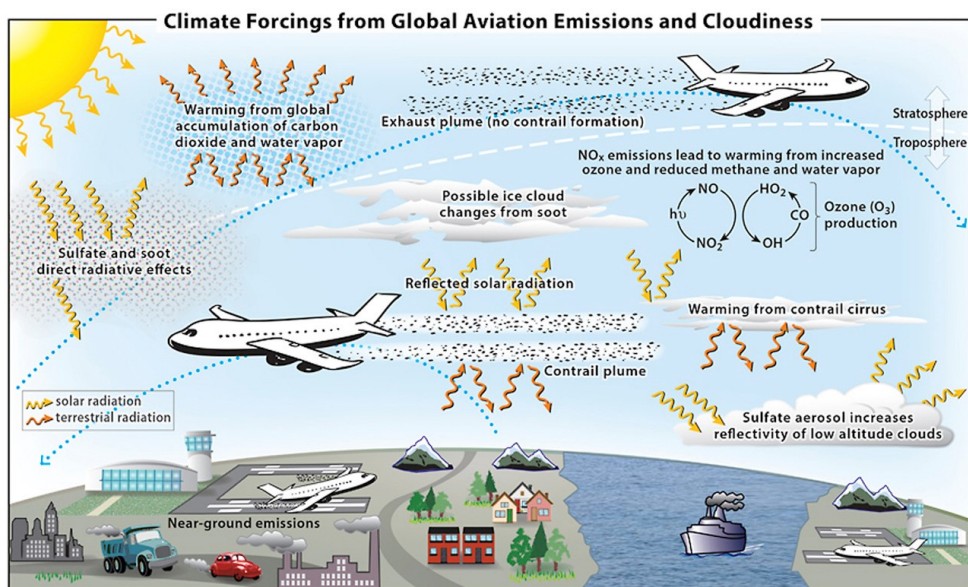

**Figure 12.** Schematic overview of climate forcing mechanisms induced by aircraft emissions. Reprinted with permission from Lee et al. [1]. Copyright 2022, Pergamon.

### 5.3. Global Climate Modelling

Quantifying aviation's global climate impact requires the use of computational climate models to predict the atmospheric response to emissions released by aircraft. Climate models simulate the climate system through mathematical representation of established physical laws (e.g., conservation of mass, energy and momentum) and a plentiful supply

of empirical data, obtained through real world observation and measurement of physical and chemical quantities (e.g., chemical concentrations, meteorological parameters) [127]. Gridded aircraft emissions inventories are used as input to climate models [77,128], in which emissions are homogeneously distributed throughout the entire grid cell they are released into, from the very instant in which they are input into the model (i.e., the ID approach). Climate models aim to simulate the chemistry, physics and dynamics of the atmosphere, along with the perturbation to the atmospheric state due to emissions, providing an output of the spatial and temporal distributions of chemical species. Radiative transfer schemes can then be used to quantify the climate response to the perturbed chemical composition, through instantaneous metrics such as RF and ERF. The variability in chemical and radiative properties of climate forcing species does however vary considerably with time. For example, $CO_2$ is distributed globally and affects the climate gradually over centuries, whereas contrails induce a severe yet short-lived, localised radiative impact [1]. Therefore, metrics such as Global Warming Potential (GWP, GWP*) and Global Temperature Potential (GTP) have been developed to better account for the temporal variation of certain properties, providing a more well-rounded analysis of the climate contributions of aircraft emission species [96,129,130].

There are two main climate model distinctions for use in aviation climate modelling: general circulation models (GCMs) and chemistry-transport models (CTMs). GCMs are highly sophisticated, yet very complex, as they are what are known as "online" models, calculating chemical composition, temperature and transport circulation simultaneously and in real time. This is often computationally expensive and may require very long processing times to run, depending on the simulation. CTMs on the other hand are reduced order "offline" models, calculating chemical composition based on pre-determined GCM results or empirically observed temperature and transport circulation data [14]. Example usage of GCMs in aviation climate modelling include the European Centre HAmburg Model (ECHAM) [131] for analysis of future contrail cirrus radiative forcing [132] and the Community Atmosphere Model version 5 (CAM5) [133] to comprehensively represent aviation aerosol climate impact [20]. CTMs appear much more frequently in the literature due to their versatility and computational efficiency. For example, the UK Meteorological Office CTM STOCHEM [134] coupled with the Common Representative Intermediates (CRI) chemical mechanism [135] is used for modelling the global impact of aviation $NO_x$ emissions on ozone production [35], MOZART CTM [136] for use in investigating the trade-off between $CO_2$ and $NO_x$ emissions [137], and another six CTMs are presented in [14] for aviation climate impact modelling purposes. Various model intercomparison studies provide an up-to-date, elaborate review of the models available for UT and LS (and thus aviation) climate modelling [138].

## 6. Nonlinear Plume-Scale Climate Effects

As detailed in the previous section, the global contributions to aviation radiative forcing have been relatively well quantified. However, much of the modelling processes used to estimate these contributions are derived from global models assuming instantaneous dilution. Therefore, any subgrid-scale chemical and physical effects that occur in the first 2–12 h upon release into the atmosphere are excluded from the model. This section reviews current scientific understanding of these plume-scale processes and sheds light on modelling methods and parametrisations that can be employed to account for nonlinear plume-scale effects.

### 6.1. Gas-Phase Photochemistry

The ultimate chemical composition of the troposphere is largely influenced by the removal of natural and anthropogenic trace species (e.g., $CH_4$, CO, $O_3$) through oxidation reactions with atmospheric free radicals. The most important oxidising species present in the troposphere is the hydroxyl radical (OH), which is relatively short-lived and occurs low concentrations owing to its rapid reactivity with atmospheric gases [139,140]. The

relative abundance of OH in the troposphere controls the degree of removal of ambient trace species, otherwise known as the atmosphere's oxidative capacity. In the purely hypothetical situation where gas-phase radical chemistry was absent and hence the oxidative capacity of the atmosphere was zero, the levels of many harmful pollutants would continue to rise unabated, resulting in a drastic change in the chemical, biological and radiative state of the Earth-atmosphere system [141]. OH is produced via atmospheric photochemistry and a process known as photolysis; a sunlight-initiated reaction of photolabile molecules to produce highly reactive species and/or radical species. Ozone is photolysed by ultraviolet light ($\lambda < 320$ nm) to produce excited singlet oxygen $O(^1D)$ (6), which then produces two hydroxyl radicals in the presence of sufficient levels of water vapour (7). The symbol M represents a non-reacting molecule in the equation, that has the role of absorbing reaction energy to create stable products.

$$O_3 + h\nu(\lambda < 320 \text{ nm}) \rightarrow O(^1D) + O_2 \tag{6}$$

$$O(^1D) + H_2O \rightarrow OH + OH \tag{7}$$

Understanding the influence of OH and the oxidative capacity of the upper troposphere is critical for the environmental analysis of aircraft emissions, as the closely-coupled chemical scheme involving hydroxyl, hydroperoxy radicals ($HO_x = OH + HO_2$) and oxides of nitrogen ($NO_x = NO + NO_2$) determines the production and loss of key climate forcing species such as ozone and methane.

Most oxidation processes that occur in the troposphere are photochemical reaction schemes involving $HO_x$, and are therefore only applicable in daylight conditions due to the reliance on solar actinic flux [37]. There is however, a range of chemical processes mainly involving nitrate radicals, that are potentially important for nighttime tropospheric oxidation processes involving key climate forcing species. See Jenkin et al. [142] for further elaboration on nighttime tropospheric chemistry.

Figure 13a from Jenkin et al. [142] is an exemplary ozone isopleth diagram that illustrates the $O_3$-$NO_x$-VOC relationship. At typical aircraft cruising altitudes (i.e., UTLS), where the emissions of $NO_x$ have a significant influence on atmospheric chemistry, it is likely that VOC content is very low relative to the background NO and $NO_2$ concentrations. Therefore, it is likely that typical VOC/$NO_x$ ratios are low in the UTLS, meaning that under the assumption of constant VOC concentrations, the $NO_x$-$O_3$ relationship takes a form similar to Figure 13b.

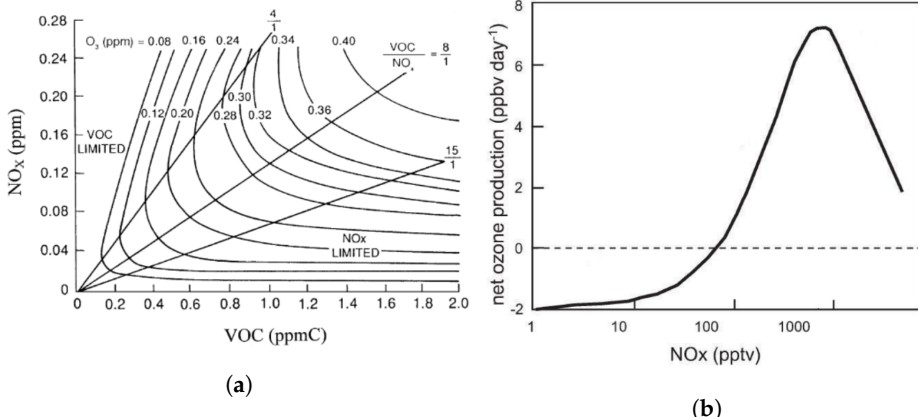

**(a)**

**(b)**

**Figure 13.** (**a**) Example isopleth diagram displaying peak ozone concentrations calculated from various initial concentrations of $NO_x$ and a specified VOC mixture, using the US EPA empirical kinetic model [143]. Reprinted with permission from Jenkin et al. [142]. Copyright 2022, Elsevier. (**b**) Variation of net ozone production efficiency with $NO_x$ concentrations, with magnitudes reflecting clean free tropospheric conditions (low VOC/$NO_x$ ratio). Reprinted with permission from Monks et al. [140]. Copyright 2022, Royal Society of Chemistry, etc.

### 6.1.1. Low-NO$_x$ Regime

In unpolluted environments characterised by low NO$_x$ concentrations, such as in regions where the ambient air is unperturbed by aircraft emissions, the dominant reaction pathway of OH is with CO (~75%), where the remaining ~25% reacts with CH$_4$ [144]. The dominant reaction pathway therefore involves the oxidation of CO to form CO$_2$, as in reaction (8) (for a detailed description of both the CO and the CH$_4$ oxidation cycles in low NO$_x$ conditions, see Wasiuk [35]). Atomic hydrogen (H) then reacts with oxygen to form HO$_2$ (9), which subsequently reacts with O$_3$ to form OH in the background troposphere (10). This initiates a chain sequence in which OH and HO$_2$ interconvert through the termination of ozone (11). This clean condition scheme therefore limits pollutant build up in the unpolluted upper troposphere and keeps ozone levels under control [145].

$$OH + CO \rightarrow H + CO_2 \tag{8}$$

$$H + O_2 + M \rightarrow HO_2 + M \tag{9}$$

$$HO_2 + O_3 \rightarrow OH + 2O_2 \tag{10}$$

$$OH + O_3 \rightarrow HO_2 + O_2 \tag{11}$$

Net reaction:

$$OH + CO + 2O_3 \rightarrow CO_2 + HO_2 + 2O_2 \tag{12}$$

Alternatively, HO$_2$ can react with itself to form hydrogen peroxide (H$_2$O$_2$) (13), or with organic peroxy radicals such as the methyl peroxy radical (CH$_3$O$_2$) to form organic hydroperoxides (14). These reaction pathways can become an effective sink for HO$_x$ under most conditions, because the formation of peroxides prevents further HO$_x$ interconversion [146].

$$HO_2 + HO_2 + M \rightarrow M + O_2 + H_2O_2 \tag{13}$$

$$CH_3O_2 + HO_2 \rightarrow CH_3O_2H + O_2 \tag{14}$$

### 6.1.2. NO$_x$-Limited Regime

In polluted environments, where NO$_x$ levels are raised considerably above ambient concentrations (e.g., typical aircraft cruising altitudes), CO oxidation takes place through reactions (8) and (9), however in the presence of nitrogen oxides, ozone is produced rather than depleted. Peroxide formation reactions (13) and (14) compete with the oxidation of NO to NO$_2$ (15) for available HO$_2$ concentrations. When the latter reaction prevails, NO$_2$ photolysis takes place, converting back to NO with ground state oxygen O($^3$P) forming as a byproduct (16). Subsequently, O($^3$P) reacts with atmospheric oxygen to produce O$_3$ (17).

$$HO_2 + NO \rightarrow OH + NO_2 \tag{15}$$

$$NO_2 + h\nu(\lambda < 420\,nm) \rightarrow NO + O(^3P) \tag{16}$$

$$O(^3P) + O_2 + M \rightarrow O_3 + M \tag{17}$$

Net reaction:

$$CO + 2O_2 \rightarrow CO_2 + O_3 \tag{18}$$

Additionally, the methane oxidation cycle leads to net ozone production in the presence of NO$_x$. The oxidation of methane by OH produces water vapour and the methyl radical (CH$_3$) (19), which further reacts with oxygen to form CH$_3$O$_2$ (20). The produced methyl peroxy radical can then react with NO to form NO$_2$ through reaction (21).

$$OH + CH_4 \rightarrow H_2O + CH_3 \tag{19}$$

$$CH_3 + O_2 + M \rightarrow M + CH_3O_2 \tag{20}$$

$$CH_3O_2 + NO \rightarrow CH_3O + NO_2 \tag{21}$$

$$CH_3O + O_2 \rightarrow HO_2 + HCHO \tag{22}$$

Net reaction:

$$CH_4 + 2O_2 \rightarrow HCHO + O_3 \tag{23}$$

The methoxy radical ($CH_3O$) produced can form additional $HO_2$ and formaldehyde ($HCHO$) through reaction (22), which is then capable of reacting to form further $NO_2$ through reaction (15). The resultant $NO_2$ produced through reactions (21) and (15) consequently produces ozone through the same pathway as CO oxidation (i.e., $NO_2$ photolysis (16) followed by reaction of the $O(^3P)$ photoproduct (17)). The net reaction of methane oxidation in polluted environments therefore results in positive ozone production. Figure 14 is a visual representation of the $NO_x$-$O_3$-CO-$CH_4$ oxidation reaction scheme described thus far.

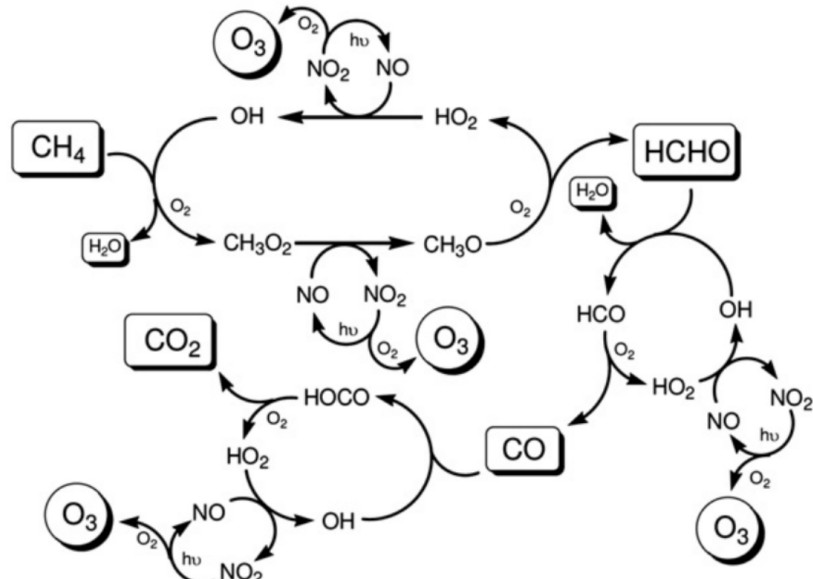

**Figure 14.** Schematic representation of the OH initiated, $NO_x$-catalysed oxidation scheme for CO and $CH_4$. Reprinted with permission from Jenkin et al. [135]. Copyright 2022, Pergamon.

6.1.3. $NO_x$-Saturated Regime

Under very high $NO_x$ conditions, such as inside aircraft exhaust plumes or in regions of the atmosphere where aircraft fly in close proximity and their corresponding emissions accumulate, the ozone production efficiency starts to drop off. This is because as NO and $NO_2$ surpass a threshold concentration, known as the compensation point (peak of the curve in Figure 13b), they begin to compete with VOCs (e.g., methane) for reaction with hydroxyl (OH), organic peroxy ($RO_2$) and peroxide $RCO_3$ radicals [35]. The products of these reactions are examples of nitrogen reservoir species: nitrous acid (HONO), nitric acid ($HNO_3$), peroxynitric acid ($HO_2NO_2$), peroxy nitrates ($RO_2NO_2$) and peroxyacyl nitrates ($RCO_3NO_2$). Reservoir species are much less efficient at forming ozone as they are more stable and are also more likely to get washed out of the atmosphere through depositional processes [142]. They are also much more stable in the upper troposphere compared to the surface equivalent [147]. Thus, reactions (24) to (28) serve as termination reactions, removing radicals from the atmosphere and preventing additional ozone formation through conversion of $NO_x$ to its more stable counterparts.

$$NO + OH + M \rightarrow HONO + M \tag{24}$$

$$NO_2 + OH + M \rightarrow HNO_3 + M \tag{25}$$

$$NO_2 + HO_2 + M \rightarrow HO_2NO_2 + M \tag{26}$$

$$NO_2 + RO_2 + M \rightarrow RO_2NO_2 + M \tag{27}$$

$$NO_2 + RCO_3 + M \rightarrow RCO_3NO_2 + M \tag{28}$$

$NO_x$ saturation effects become particularly important to observe in the case of diluting aircraft exhaust plumes, because the high $NO_X$-VOC ratio means that termination reactions are often favoured over catalytic ozone formation [148]. In the relatively fresh exhaust plume (first 10 min) where $NO_x$ concentrations are significantly enhanced, ozone titration by NO results in large scale production of $NO_2$ (29). Additionally, it decreases the formation of $HO_x$, due to depleted ozone levels in the plume.

$$NO + O_3 \rightarrow NO_2 + O_2 \tag{29}$$

The dilution of the plume results in reduced $NO_x$ concentrations over time, and the in-plume chemistry transitions from $NO_x$-saturated to $NO_x$-limited. With ozone levels still depleted, the remaining $HO_2$ and $RO_2$ in the plume (formed from the oxidation of CO and VOCs by OH) react with the remaining NO to produce OH and $NO_2$ without further depleting ozone. This leads to increasing OH and $NO_2$ concentrations which give rise to a net ozone recovery due to $NO_2$ photolysis (reactions (16) and (17)), with full recovery to ambient concentrations within 1–2 h post emission. As ozone levels rise in the plume back towards ambient concentrations, the photochemical formation of OH becomes more common, through reactions (6) and (7). Newly formed OH can oxidise CO and VOCs to form peroxy radicals which catalyse ozone production, however it can also react with NO and $NO_2$ to form stable nitrogen reservoir species, through reactions (24) to (28) [50].

In the ID scenario inherent to large-scale climate models, ozone titration doesn't occur on the same scale due to lower mixing ratios of $NO_x$ when instantly diluted. Therefore, initial ozone depletion and subsequent recovery in the exhaust plume is not properly captured, meaning that instead, ozone levels remain reasonably high throughout and hence $HO_x$ production can remain stable. The stable $HO_x$ levels mean $NO_x$ to reservoir species conversion remains stable also, throughout the $NO_x$ lifetime. On a global scale, it has been shown that inclusion of plume processes in modelling efforts leads to a net reduction in ozone forming potential of aviation $NO_x$ emissions. Vohralik et al. [48] summarises the estimates made for the degree of reduction in ozone forming potential when plume effects were included. Initial findings from Kraabøl et al. [149] and Meijer et al. [150] estimated discrepancies in ozone production of 15–18%, however these studies only accounted for ozone depletion in the plume, and not the $O_3$ generated during plume expansion. Inclusion of both ozone depletion and production in the plume in Meijer [51], led to updated estimates in ozone formation changes 0% to −5% in January and +5% to −10% in July, indicating that plume processing can actually increase net ozone production when propagated to global scales.

### 6.2. Heterogeneous Chemistry

Reactions occurring in the atmosphere on either the gas–solid interface (e.g., aerosol particles) or the gas–liquid interface (e.g., cloud droplets) are referred to as heterogeneous reactions. The heterogeneous chemistry which can affect ozone concentrations through production and loss of $HO_x$ and $NO_x$ and the production of halogen radicals is extremely important in particle rich aircraft exhaust plumes and contrails [151]. The exhaust plume contains emitted soot particles and ultrafine aqueous aerosol particles which are either formed within the plume or entrained into the plume from ambient air, as elaborated on in the following subsection. In the case of contrail formation, heterogeneous chemistry becomes very efficient, due to the four-fold increase in particle surface area of contrail ice compared to typical exhaust and background aerosol surface area. Meilinger et al. [152] states that the heterogeneous reactions occurring on aerosol particles have a negligible effect

on ozone, however contrail ice can influence the ozone response to aircraft emissions by $\pm 0.5\%$ on a macroscopic scale which varies depending on time of day and year. Due to the relatively minor impact heterogeneous chemical reactions have on aircraft-induced ozone perturbations, and hence on aviation climate impact, their effect will be acknowledged, but the chemical intricacies will not be discussed further. For more information, see references contained within this paragraph.

### 6.3. Aerosol and Contrail Microphysics

Whilst the climate impact of aviation $NO_x$ emissions is dependent on the photochemical processes catalysing ozone production and methane destruction, the evolution and radiative forcing of aerosols and contrails is predominantly controlled by microphysical processes that occur at the aircraft plume scale.

#### 6.3.1. The Microphysical Formation of Aerosols and Contrails

Upon release into the atmosphere, aerosol particle formation occurs through one of two nucleation pathways:

1.  The condensation of two distinct gas phase molecules to form a liquid phase droplet through what is known as binary homogeneous nucleation. This is the case for reactive sulphur emissions that get chemically oxidised into sulphuric acid ($H_2SO_4$), which then condenses with water vapour to form $H_2SO_4$ / $H_2O$ droplets [153].
2.  The gas-to-particle conversion occurring on the surface of foreign particles is known as binary heterogeneous nucleation, often leading to a liquid coating that forms on the particle. For example, $H_2SO_4$ / $H_2O$ droplets can form a partial liquid coating around chemically activated soot in the aircraft exhaust plume through heterogeneous nucleation, leading to soot aerosol formation; a process that plays an important role in the formation of contrails [154].

Newly formed aerosol particles subsequently grow in the aircraft wake due to further condensation by uptake of surrounding water vapour, and a process known as coagulation. Coagulation refers to the collision between particles that results in the formation of larger particles, often initiated by graviational settling, turbulence or thermal motion. A process called scavenging occurs when two particles, one much larger than the other, coagulate and lead to the removal of the small particle from its particular size category, with its mass contributing slightly to the increase in mass of the larger particle. Coagulation commonly occurs in aircraft wakes between volatile particles such as $H_2SO_4$ / $H_2O$ particles and soot aerosol, forming a mixed $H_2SO_4$ / $H_2O$-soot aerosol (see Figure 15).

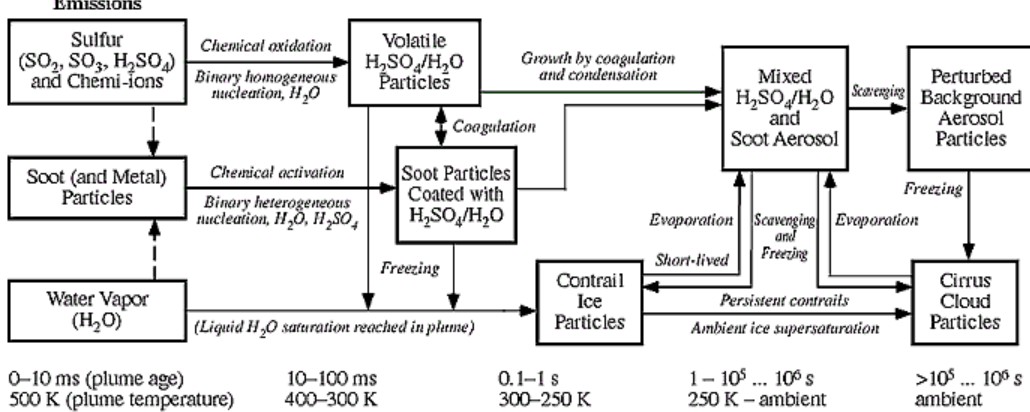

**Figure 15.** The microphysical formation of aerosol and contrail particles in an aircraft plume. Displayed as a function of plume age and temperature. Reprinted with permission from IPCC special report (1999) [14]. Copyright 2022, Cambridge University Press.

Sulphate-soot aerosol may eventually become scavenged by background aerosol particles if it remains stable for up to a day [14]. Alternatively, if at any point during the mixing process of aircraft exhaust with air, water-supersaturation is exceeded, aircraft-induced particles and entrained background aerosol can be activated into water droplets through the uptake of surrounding water vapour [100]. If temperatures are below threshold according to the SA criterion, then these aerosol-activated water droplets will freeze to form contrail ice particles in the aircraft wake on the order of seconds, post emission.

6.3.2. Contrail Microphysical Properties

The aforementioned microphysical processes (i.e., nucleation, condensation, coagulation, scavenging and freezing) determine the eventual composition and size distribution of particles in the aircraft wake, and if thermodynamic conditions permit, the formation and evolution of contrail ice particles [32,111]. Hereinafter, this section will focus on the microphysics of contrail ice particles, as they induce the most significant radiative response out of all aviation climate forcers. Findings from Kärcher et al. [155] conclude that the predominant ice nucleation pathway for contrail formation is heterogeneous freezing of chemically activated soot aerosol particles, as these are the only remaining particle type in sufficient abundance at the time of freezing. However, simulation results from Kärcher et al. [83] strongly suggest that contrail formation is still likely in the absence of soot and sulphur emissions, through the activation and freezing of background aerosols.

The radiative forcing of contrails is thought to be determined by the product of optical depth and areal coverage [106]. In terms of areal coverage, contrail forcing is thus dominated by the presence of persistent contrails and contrail cirrus. However, the optical depth of a contrail is less dependent on its macroscopic properties and is instead determined by the optical and physical characteristics of the ice particles on the microscopic scale. The microphysical parameters deemed to be most responsible for inducing a radiative response are ice water content (amount of cloud ice per unit volume [100]), total ice particle number concentration, ice particle size distributions, effective radii, and ice particle shape [156]. Various studies have formulated methods to estimate RF and ERF from contrails, based on the parametrisation of these properties [157–159]. It is defined that the optical depth is proportional to the ice water content divided by the effective radius of entrained ice particles [158]. Contrails that tend to have more aspherical ice particle shapes are likely to have a stronger solar albedo, increasing the reflectance of the SW flux [157]. Ice number concentration is also thought to increase optical depth [160].

Optical depth of contrail cirrus is generally seen to increase radiative forcing, however the relationship is nonlinear and under assumed atmospheric conditions [157,161], contrail RF increases with optical depth up to around 2.0, where it peaks, before decreasing at higher optical depths (see Figure 16a). This is due to the lessening increase of positive LW forcing with increasing optical depth, whilst negative SW forcing continues to drop at a similar rate throughout. In addition to contrail optical properties, the magnitude and direction of contrail forcing is also directly determined by solar position, otherwise known as the solar zenith angle (SZA) (see Figure 16b). Throughout the range of SZA values, LW forcing remains constant, because infrared emission from the Earth's surface is unaffected by solar flux. The SW flux on the other hand, goes further negative as SZA increases, up until a maximum at around 70°. Beyond this, the Sun begins to set and SW forcing returns to zero when the Sun is at 90° to the Earth's surface. This confirms the notion that contrails are solely warming at night, as the positive LW forcing always remains constant, whilst at night there is no chance of negative SW forcing.

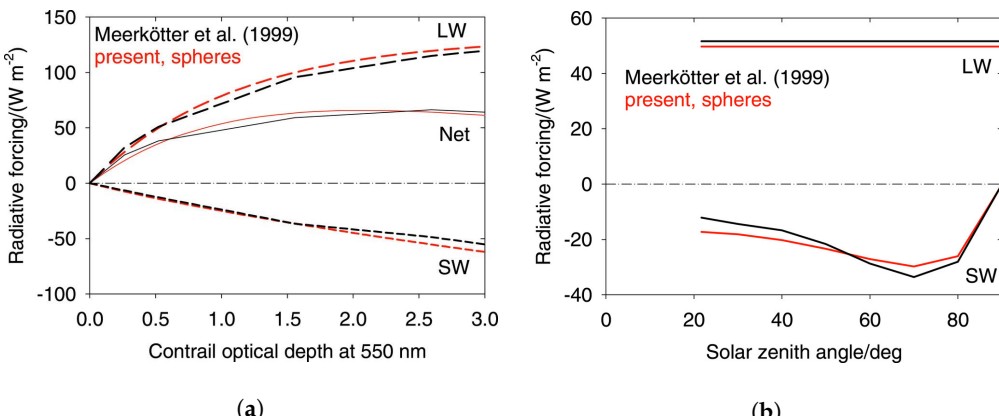

(**a**)                                                                    (**b**)

**Figure 16.** (**a**) Daily mean instantaneous SW, LW and net components of RF assuming 100% contrail cover vs optical depth as calculated using the spherical ice particle model from Schumann et al. [161]. Results from Meerkötter et al. [157] shown for comparison. (**b**) RF vs SZA for spherical ice particles, with Meerkötter comparison shown. Reprinted with permission from [161]. Copyright 2022, American Meteorological Society.

### 6.4. The Saturation of Aircraft Emissions in High-Density Airspace Regions

In dense airspace regions, where the frequency of traversing aircraft is high, the resulting exhaust plumes may intersect and overlap, further affecting the nonlinear atmospheric response to chemical and microphysical processing occurring at the plume scale. Two outstanding saturation effects documented in the literature include the surpassing of $NO_x$-saturated conditions and the dehydration of surrounding water vapour due to contrail formation, leading to the mutual inhibition of ice particle growth in the aircraft wake.

#### 6.4.1. $NO_x$-Saturated Conditions

The accumulation of nitrogen oxide emissions in the troposphere due to overlapping aircraft plumes has been observed empirically. For example, Schlager et al. [54] witnessed $NO_x$ concentrations of up to 30 times the average background concentrations in the NAFC, for an overlap of 2 to 5 aircraft plumes. As discussed in Section 6.1, the net ozone production rate in the upper troposphere is a nonlinear function of the concentrations of NO and $NO_2$. This means that increases in $NO_x$ lead to increases in $O_3$ production, up until a maximum where any additional $NO_x$ results in reduced ozone production efficiency (e.g., Figure 13b). The turnover point, sometimes called the "compensation point" is determined by competitive reactions involving nitrogen species and $HO_x$. In the $NO_x$-limited regime (left of the minimum ozone production efficiency $P(O_3)_{max}$), NO drives the production of ozone through reaction with the hydroperoxy radical ($HO_2$) [140]. However, it also drives the removal of $HO_x$ through the reaction of OH with $HO_2$, $HNO_4$ and $NO_2$, thus limiting $HO_x$ available for further $O_3$ production [162]. As $NO_x$ levels increase up to the compensation point, the increasing competition of $HO_x$ removal processes begins to level off ozone production efficiency. Beyond this point, further increases in $NO_x$ serve to decrease $P(O_3)$, thus signalling the start of the $NO_x$-saturated regime.

During the POLINAT/SONEX campaign, Jaeglé et al. [5] presented the first empirical evidence of $NO_x$-saturated conditions in the NAFC. As seen in Figure 17, $P(O_3)$ levels increased with increasing $NO_x$ up until the saturation point, beyond which ozone production begins to drop off.

As seen in figure 17 at 100–200 pptv/day of $HO_x$ production, $P(O_3)$ levels increased with increasing $NO_x$ up until around 300 pptv, where it is clear that $NO_x$ saturation has been reached. This means that any further emission of $NO_x$ under those specific atmospheric conditions will potentially serve to decrease the net ozone production rate. Generally speaking, ozone production efficiency, and hence the threshold for $NO_x$-saturated conditions, is largely dependent on just two variables: the ambient $NO_x$ concentration (including the

accumulation of $NO_x$ from lingering aircraft plumes) and the $HO_x$ production rate (which depends on the chemical composition of the atmosphere and the solar intensity [140]) [163].

Furthermore, model results from Kraabøl et al. [112] in a study observing interactions between plumes revealed that, for aircraft flying along the same track, plumes of follower aircraft exhibit a considerably smaller ozone response than that of the leader aircraft. This observation on a local scale opens up the discussion for controlled saturation of emissions through formation flight, to take advantage of effects that are beneficial to the climate.

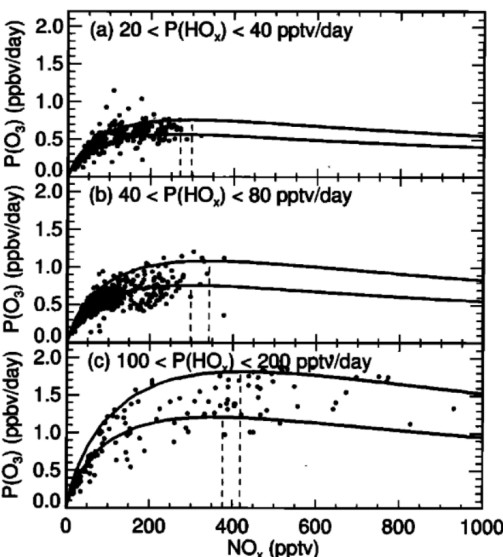

**Figure 17.** Empirical observations of ozone production rate as a function of $NO_x$ concentrations (parts per trillion volume, pptv), for three ranges of primary $HO_x$ production: (**a**) 20–40 pptv/day, (**b**) 40–80 pptv/day, and (**c**) 100–200 pptv/day. Solid lines represent the range limits from model calculations. Dashed lines depict the $NO_x$ level that induces peak ozone response. Reprinted with permission from Jaeglé et al. [5]. Copyright 2022, American Geophysical Union.

6.4.2. Dehydration Effects due to Contrail Formation

Contrails grow and persist in the atmosphere through deposition of water vapour from ice-supersaturated layers of the atmosphere. The conversion of atmospheric water vapour to solid phase ice particles has the potential to deplete local water vapour concentrations in regions conducive to persistent contrail generation [164]. It is thought that the depositional growth of contrail cirrus in regions of high airspace density can even lead to the dehydration of $H_2O$ at typical flight levels, followed by the redistribution of humidity to lower levels, due to sedimentation and advection processes [6].

Contrail dehydration effects were first quantified by Burkhardt and Kärcher [105], in which a global atmospheric model was used to perform long-term integrations of the global impact of contrail cirrus on natural cirrus reduction due to ambient water vapour depletion. The paper found initial RF estimates to be $-7$ mWm$^{-2}$, lessening the contrail cirrus RF estimate by a factor of approximately one fifth, however the results were not conclusive. Schumann et al. [6] furthered this investigation through quantification of the impact of water exchange on contrail properties, large-scale humidity and the background climate. The results suggested that the drying at flight levels caused contrails to become thinner and last longer in the atmosphere, and the net reduction in contrail cirrus RF was deemed to be ~15% (not too dissimilar to the results of [105]). Furthermore, the results from a second run of the model, with aircraft emissions enhanced by 100 times, showing increasingly significant dehydration effects due to contrail cirrus formation and the redirection of humidity to lower flight levels. This implies that local dehydration effects will be far more significant in dense airspace regions where emissions accumulate and contrails can overlap, such as over mainland areas and in flight corridors. Work by Unterstrasser [164,165] has

elaborated on this concept of local dehydration, through numerical analysis of contrail growth in close proximity flight scenarios. The studies conclude that aircraft contrails compete for available water vapour, mutually inhibiting growth and leading to a saturation effect which diminishes the properties of contrails formed by subsequent aircraft travelling through that region. Under proposed formation flight scenarios, the total extinction (optical depth) and total ice mass behind a two-aircraft formation are found to be reduced by 20–50% and 30–60% respectively.

The presence of plume-scale effects that are amplified in close-proximity flight scenarios begs the question; can climate beneficial saturation effects such as $NO_x$-saturated regimes and contrail-induced dehydration be exploited for mitigation purposes? Section 7 explores the concept of formation flight and how these additional climate benefits from emissions saturation may provide further impetus for real-world implementation.

### 6.5. Parametrisation of Plume-Scale Effects into Global Models

Plume modelling methods serve as a useful tool for analysing the chemical and physical evolution of aircraft exhaust plumes throughout their lifetime. However, integrating high-resolution plume data into global atmospheric models is simply unfeasible when a large number of flights, on the order of 100,000–200,000 flights per day [166], must be considered. To counteract this, simpler parametrisations of plume models are used, which capture so-called "effective emissions". These parametrisations attempt to correct the ID response of the large-scale model, according to the predicted plume-scale effects and their influence on the eventual climate impact. Parametrising plume-scale climate effects into global atmospheric models that assume instantaneous dispersion is a topic covered extensively in the literature, for both ozone-$NO_x$ chemistry [4,39,150,167] and contrail and cloud processes [168].

#### 6.5.1. Parametrisation of Gas-Phase Chemical Conversions

In Paoli et al. [4], three key concepts are reviewed which enable the parametrisation of nonlinear plume chemistry into ID global models; effective emission indices (EEIs), effective conversion factors (ECFs) and effective reaction rates (ERRs).

The EEI concept was first theorised in Petry et al. [38], in an attempt to account for the difference in concentration evolution of key chemical species between large scale models which assume ID and plume models which account for the entrainment of emissions throughout the plume lifetime. EEIs provide a suitable correction to the original EI of an emitted species, so that the concentration is the same in both models at the end of the plume "dispersion time" $t_{ref}$, when emissions are fully dispersed into the dimensions of the computational grid cell in which they were released.

Meijer [51] presents the concept of ECFs which are factors applied to the $NO_x$ emission index to account for the increased chemical conversion rate of $NO_x$ to nitrogen reservoir species in the plume. Since the total emitted reactive nitrogen ($NO_y$) is chemically conserved throughout the plume lifetime, the amount of $NO_y$ throughout the plume lifetime is equal to the amount of $NO_x$ emitted initially, and hence the sum of ECFs for $NO_x$ and all reservoir species is unity [48]. Thus, the faster in-plume conversion rates lead to a decreased $NO_x$ ECF, whilst the ECFs of reservoir species such as $HNO_2$ and $HNO_3$ increase considerably. As a result, the eventual net ozone production is affected as explained in Section 6.1, however ECFs vary considerably depending on altitude, latitude and seasonal variation meaning the magnitude and direction of the ozone perturbation is also affected.

Despite the widespread use of EEIs and ECFs to parametrise plume-scale gas-phase chemical conversions in past literature, there are a number of known issues that affect accuracy such as mass conservation inconsistencies and poor accounting of local and regional variation in dispersion properties and turbulence. Cariolle et al. [39] attempts to overcome such issues with the ERR concept. ERRs reconstruct the concentrations of the emitted species in the plume by diluting the emission to the resolution of the model's computational grid cells, while modified reaction rates are used to model reactions

with ambient chemical species that occur at the plume scale. Modified reaction rates are introduced by the determination of effective reaction rate constants, which are used to compute secondary species produced in the plume. As a result, ERRs build on the EEI and ECF concepts by ensuring full mass conservation through modulation of pre-existing background chemical cycles as opposed to direct estimation of mass changes (as is performed in EEI and ECF parametrisations). This method therefore accounts for chemical transport due to diffusion and turbulence processes that are entirely dependent on location, increasing accuracy under a wide range of atmospheric conditions.

6.5.2. Parametrisation of Heterogeneous Chemistry and Microphysics

Despite there being an extensive number of gas-phase chemistry parametrisations, the same cannot be said for parametrisations of heterogeneous chemistry and microphysics when modelling aircraft plume-scale climate effects at a global scale. Kärcher et al. [83] was one of the first studies to explicitly attempt to parametrise heterogeneous reactions and aerosol microphysics in global models. The paper attributed perturbations to background aerosol layer concentrations (due to aerosol emissions from aircraft) to increasing surface areas and number concentrations of aerosol particles, which subsequently increase heterogeneous reaction rates. Findings from Meilinger et al. [152,169] however, reveal that heterogeneous chemistry effects in dispersing aircraft plumes require special consideration of both the chemical and microphysical interactions, and the dynamical response of the plume itself. In [152], the Mainz Aircraft Plume Model is used to calculate the response of ozone and nitrogen reservoir species in the plume due to heterogeneous chemistry and microphysical effects. It is found from the modelling results that the impact of heterogeneous chemistry and microphysics on the ozone response at global scales is highly sensitive to local meteorology and the instantaneous state of the atmosphere. Therefore, parametrisation of these effects is much more convoluted than first thought, and due to the relatively minute impact on ozone response ($\pm 0.5\%$), it is not unreasonable to disregard these effects in global modelling efforts.

Contrarily, the parametrisation of contrail microphysics into global models is of crucial importance, as contrail radiative forcing is determined primarily by the optical properties of its consituent ice particles. Burkhardt and Kärcher [168] summarise the parametrisations necessary to include contrail radiative forcing in global models; this includes parametrising the factors affecting ice supersaturation, contrail formation and persistence, contrail spreading and ice water content. This led to the development of the process-based contrail cirrus module (CCMod), which was later implemented in the global climate model ECHAM4, for global contrail cirrus radiative forcing analysis [105,170,171]. More recently, a microphysical extension to CCMod was applied and implemented in ECHAM5 [159,172]. Model outputs were used to determine the global effective radiative forcing of contrail cirrus in [1], finding that the ERF is more than 50% smaller than the RF equivalent. This is thought to be due to the reduction in natural cloudiness caused by contrail cirrus dehydration of the surrounding atmosphere, and due to the dependence of contrail climate impact on prevailing air traffic distribution patterns. See [1] for more info on the parametrisations of contrail and aerosol microphysics implemented to deduce global ERF estimates.

## 7. Aviation Climate Impact Mitigation

It is customary practice among aviation policymakers to solely focus on mitigating the greenhouse effect induced by aviation carbon emissions. This means that analysis of the climate impact of non-$CO_2$ emissions is largely neglected, despite being responsible for over two-thirds of aviation-induced climate change [1].

### 7.1. Conventional $CO_2$-Centric Mitigation Approach

The industry fixation on aviation $CO_2$ mitigation stems from the easily quantifiable nature of the substance and its impacts; the direct coupling to fuel consumption and its relative stability and long lifetime compared to other aviation emission species make

it a prime target for mitigation. This is due to the mutually assured reduction in both fuel consumption and corresponding $CO_2$ emissions, providing both an economic and environmental incentive. Conventional mitigation approaches have often taken the form of improvements to aircraft and engine design, aviation technology and infrastructure, and aircraft operations, all aimed at minimising aircraft fuel consumption. These changes have been largely incremental due to the prioritisation of safety and economic stability over rapid implementation. As a result, efficiency gains have stalled in recent years (1–2% per annum), whilst growth rates continue to rise unabated (4–5% per annum) [1,173]. See Figure 18a for a graphical representation of the trend in efficiency gains and aviation growth since 1940.

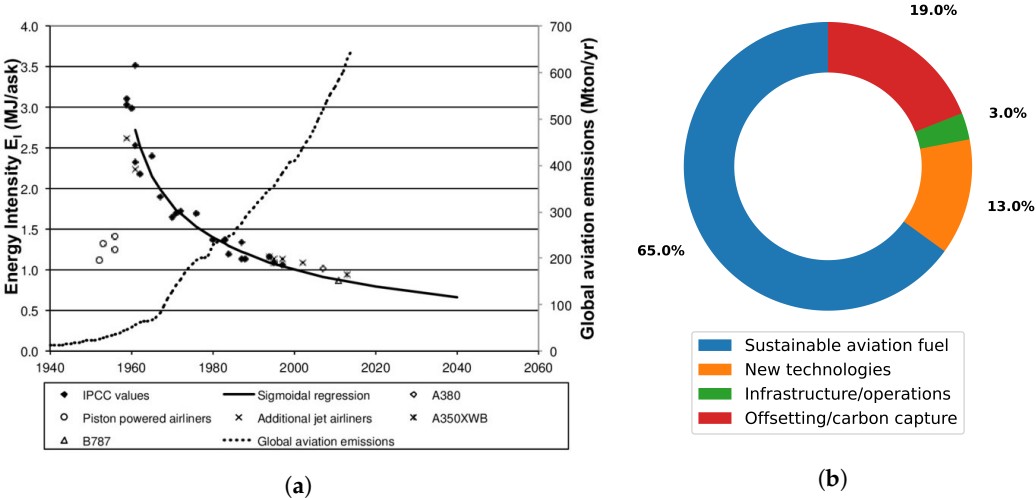

(a)　　　　　　　　　　　　　　　　　　　　　　(b)

**Figure 18.** (**a**) Long haul aircraft efficiency gains versus global aviation emissions growth since 1940 and projected to 2040. Reprinted with permission from Peeters et al. [173]. Copyright 2022, Pergamon. (**b**) IATA Net Zero 2050 contributions. Data derived from [174].

At the International Air Transport Association (IATA) 77th Annual General Meeting in 2021, a resolution was approved for the global air transport industry to achieve net zero carbon emissions by 2050, in accordance with the Paris climate agreement to limit global temperature rise to 1.5 °C [174]. This plan relies on the following contributions, as displayed in Figure 18b: sustainable aviation fuels (SAFs) are to provide 65% of the net carbon reduction, offsetting schemes and carbon capture are to deliver 19%, new technologies such as hydrogen and electric propulsion to provide 13%, and infrastructure/operational improvements to provide the final 3%. This means that 78% of the prospective net carbon emissions reduction comes from SAFs and alternative technology such as hydrogen and electric propulsion concepts.

The fundamental issue with reliance on SAFs and nascent technologies, is that they may take decades to bring about noteworthy reductions in aviation climate impact. This is due to the need for a fleet-wide overhaul of aircraft equipment and associated ground infrastructure, which would likely incur huge costs and require extensive certification and approval from aviation authoritative bodies [175]. Furthermore, environmental and ethical concerns have cast doubts over the prospect of a 4000 fold increase in global SAF production, to meet the IATA net zero target of 449 billion litres per year by 2050 [174,176].

Carbon offsetting schemes, such as ICAO's Carbon Offsetting and Reduction Scheme for International Aviation (CORSIA), have been proposed to alleviate aviation climate impact in the meantime. However, an EU study investigating the efficacy of carbon offsetting indicates that such schemes often fall far short on providing meaningful mitigation, because of questionable offset quality, perverse incentives taking away from the real need to decarbonise, a lack of participation from key markets, and a clear lack of overall transparency and enforceability [177,178].

The remaining net zero contribution is expected to come from improvements to operations and infrastructure, which will continually provide 1–2% per year. This will involve

transitioning towards more efficient fuel management systems and improving overall technology and design of the aircraft, as well as advancing air traffic management systems and airspace modernisation [173]. However, as mentioned previously, improvements aimed at increasing aircraft fuel efficiency are largely outpaced by growth rates in passenger demand and hence emissions. Therefore, more must be done in the short term (i.e., the next 5 to 10 years), to reduce the dependency on nascent technologies which are still decades away from being impactful.

Whilst the proposed net zero targets are solely focused on decarbonising the aviation sector, it is inherently assumed that the reduction in non-$CO_2$ emissions will follow a similar trend. This review paper has evidenced however that this is not necessarily the case, due to the sensitivity of non-$CO_2$ emissions to combustor conditions, as well as the dependency of the atmospheric response on the environmental conditions of the ambient air. For this reason, action must be taken simultaneously to tackle non-$CO_2$ climate impact, alongside the current decarbonisation plan. This brings with it the potential to provide substantial climate impact reductions on a much shorter timescale than current net zero strategies.

*7.2. Alternative Non-CO₂-Focused Mitigation Approach*

There are myriad solutions to mitigating aviation climate impact in the interim to low-carbon flight, however the focus here will remain on near-term operational mitigation measures, that optimise aircraft routing by minimising non carbon-based radiative effects.

As alluded to in Sections 2 and 5, non-$CO_2$ emissions are on the other hand, produced in varying quantities depending on combustor conditions. Also, when released into the atmosphere, they have a spatio-temporally sensitive climate response, which means that the net RF varies depending on the background atmospheric conditions into which these species are emitted. As such, anthropogenic as well as natural perturbations to atmospheric chemistry need to be considered; the entrainment of emissions to the exhaust plume for several hours post emission, leads to locally elevated emissions concentrations. This causes reactive non-$CO_2$ species to experience nonlinear chemical and microphysical processing that occurs at the scale of the plume, which subsequently affects their net climatic response when propagated to global scales.

Measures such as climate-optimal aircraft routing and formation flight present the potential for a substantial climate impact reduction in a relatively short timeframe, whilst requiring minimal technology changes to the current fleet and ground infrastructure.

### 7.2.1. Climate-Optimal Aircraft Routing

The first measure proposed to tackle aviation's non carbon-based environmental impact is climate-optimal aircraft routing. This involves re-routing aircraft in flight to avoid regions of the atmosphere that are particularly sensitive to non-$CO_2$ climatic effects, such as where $NO_x$ gives rise to excessive ozone production or where persistent contrails are formed. Simulation efforts have suggested that this method has the potential to reduce aviation climate impact by 10–20%, at the cost of only a few percent of additional fuel consumption [7,179,180]. Moreover, the non-$CO_2$ climate impact from aviation is not evenly distributed across all flight distances, meaning that operational mitigation can be targeted towards flights which induce the most significant impact. For example, in a case study on contrail avoidance strategy [181], it was found that diverting 1.7% of the aircraft fleet could reduce the contrail climate impact by 59.3%, with only a 0.014% increase in fuel burn and accompanying additional $CO_2$ emissions. This sheds light on the sheer potential for significant reduction in aviation climate impact, through a minimally invasive climate-focused route optimisation strategy. Figure 19 illustrates how the cumulative contrail climate impact varies with proportion of flights, highlighting that only ~2% of flights are responsible for ~80% of the contrail energy forcing.

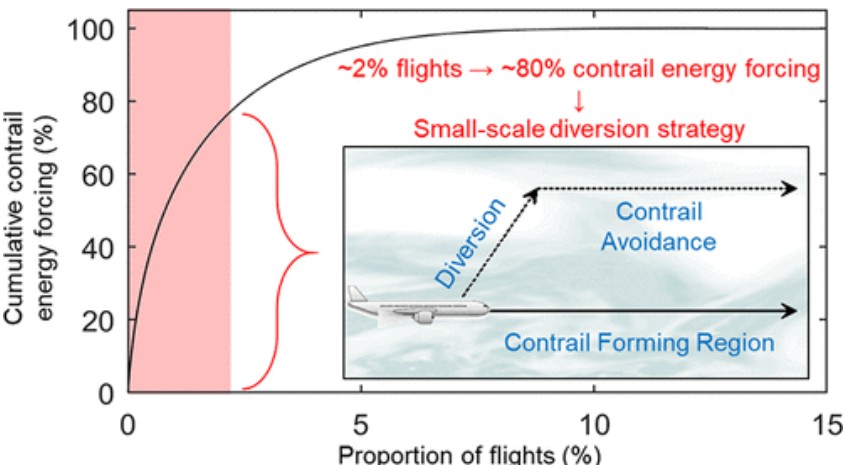

**Figure 19.** Schematic representation of contrail avoidance strategy. Plot of cumulative contrail energy forcing against proportion of flights. Reprinted with permission from [181]. Copyright 2022, American Chemical Society.

Implementation of climate-optimal routing procedures in the real world do however require a number of issues to be firstly be addressed; Grewe et al. [182] lay out four key hurdles that must be overcome:

1. The accuracy and robustness in determining eco-efficient flight trajectories must be improved, and this must be possible near real time.
2. Consensus must be achieved on determining the extent to which cooling effects should be exploited (e.g., intentionally flying a route that generates a cooling contrail could be seen as unnecessary intervention in nature).
3. The implications of fleet-wide climate-optimal routing on the air traffic management system must be assessed rigorously, to ensure safety, order and efficiency is maintained.
4. A non-$CO_2$ market-based measure or policy pathway must be adopted to incentivise this transition towards a climate-optimised air traffic network.

In the time since, a dedicated network of researchers from various institutions have been working towards bringing this concept to fruition on a large scale. The FlyATM4E research group has been developing methodologies to increase robustness in the determination of eco-efficient flight trajectories using so-called algorithmic climate change functions (aCCFs) [183]. Live trials are currently ongoing in the Maastricht Upper Area Control (MUAC) to investigate the operational feasibility of contrail prevention from an air traffic control perspective [184]. The inclusion of non-$CO_2$ effects of aviation in the European Union emissions trading scheme (EU ETS) and under CORSIA has also been discussed in great detail in a comprehensive report by Niklaß et al. [185]. Most recently, a comprehensive survey paper was published by Simorgh et al. [186], detailing the current operational strategies that are proposed in the state-of-the-art literature published in this field over the last few decades. The aim of this review was to collate methodologies used for aircraft performance modelling, climate modelling and optimisation, and to identify gaps to guide future research.

### 7.2.2. Formation Flight

Formation flight for wake energy retrieval involves the flight of two or more aircraft, with the follower aircraft positioned in the smooth updraft of the leader aircraft's wake. This has been demonstrated in simulation [8,187] and flight testing [188] to reduce required lift and thrust of the trailing aircraft and offer a 5–10% reduction in fuel burn in paired formation. Most work to date focuses on two-aircraft configurations, however three [189] and more have also been considered. In Figure 20, it is evident how this concept is used in practice to obtain aerodynamic benefits. The counter-rotation of the wake vortices of the leader aircraft leads to a stream of upwash on the outside of either rolled up vortex. Flying

a follower aircraft in this region of upwash induces a vertical velocity, with aerodynamic benefits achieved at distances as much as 30 wingspans (i.e., ~1 km for Airbus A320).

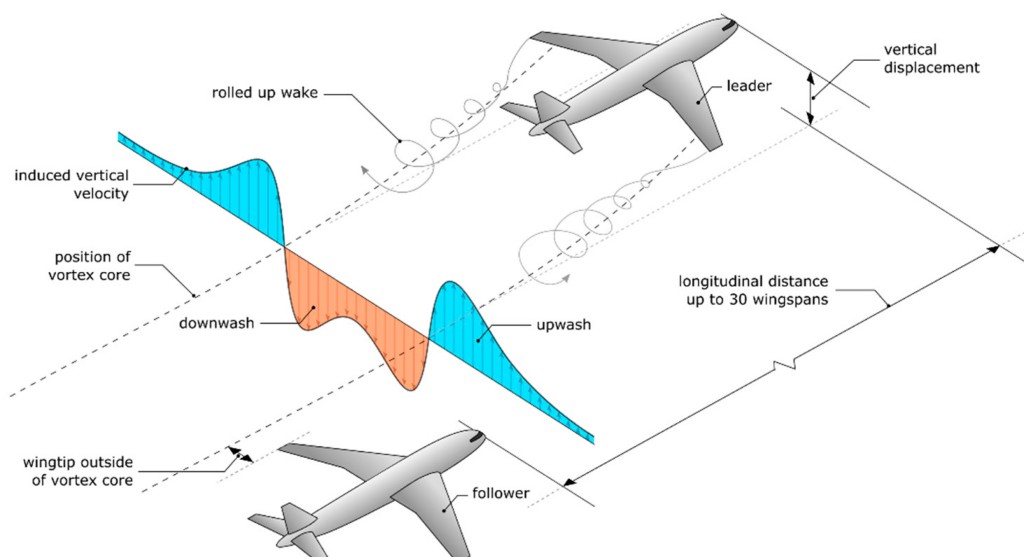

**Figure 20.** Visual representation of the formation flight concept, depicting the fundamental principles of wake energy retrieval [190]. Figure obtained from Marks et al. [190] licensed under CC BY 4.0.

Safety is a critical hurdle, particularly in close formation cases, along with regulatory and air traffic systems and processes. Airbus' fello'fly wake energy retrieval project has undertaken initial work to identify and address operational and safety challenges, proposing adaptations to regulatory standards to facilitate adoption, and conducting a trial transatlantic flight in 2021 [191]. Claims in this document state that formation flight strategies could be deployed around the middle of this decade. Proposed longitudinal separation distances for formation flight range from 56 ft [188] to 1.5 NM [191], with lateral distances ranging from 10%-span overlap to 30 m separation showing benefit. Routing aircraft into formation has been considered in simulation [192], with net fuel (and hence $CO_2$ saving benefits of 5.8% being identified for a single operator, rising to 7.7% for a transatlantic alliance. Significant benefits have been projected for long-haul and transatlantic routes, with more modest reductions for low-cost airline cases [9].

Non-$CO_2$ impacts have been neglected until recently, but a synergistic outcome of formation flight is realised via the saturation of emissions in the trailing exhaust plumes, leading to the aforementioned nonlinear chemical and microphysical response that reduces ozone and contrail formation, as detailed in Section 6.4. This has been projected to offer net climate impact reduction for a two-aircraft formation of 13–33% [10], which is significantly higher than the $CO_2$-focused formation discount factors employed in the routing simulations mentioned above. Given the near-term feasibility demonstrated in industry-led trials, and the combined $CO_2$ and non-$CO_2$ benefits that produce an outsized impact when overall ERF is considered, formation flight offers a promising interim solution that can continue to be employed for drag reduction/wake energy retrieval once SAFs and hydrogen/electric propulsion systems are more widely used.

In this section, the prospects of non carbon-based operational mitigation measures have been discussed, and their potential to reduce aviation's net climate impact has been investigated. Gradual efficiency gains are unable to outpace the rapid growth in demand year upon year, meaning that more radical approaches must be implemented in order to achieve net zero emissions targets. For this reason, two concepts have been proposed and discussed: climate-optimal aircraft routing and formation flight. The former aims to re-route the aircraft to minimise time spent in particularly climate sensitive regions of the atmosphere, whereas the latter aims to optimise atmospheric conditions artificially by overlapping aircraft plumes and saturating local concentrations of emissions to lead

to climate-beneficial effects. For the industry to truly reap the benefits and maximise potential non-$CO_2$ reductions from implementation of these measures, there is the potential to perform them simultaneously, in a controlled and safe manner. This would involve flying aircraft in formation, along routes which are optimised with respect to minimum climate impact.

## 8. Conclusions

This review paper delivers a holistic overview of literature pertaining to the spatio-temporal climate sensitivity of the atmosphere to reactive non carbon-based species. This includes literature from a range of research areas: aircraft emissions, plume dispersion and dynamics, the distribution of air traffic and corresponding emissions, the climate impact of aviation on both a global and a local scale, and lastly the mitigation potential for both $CO_2$ and non-$CO_2$ climate impact.

A strong finding from this review, giving direction to fruitful future work on this research topic, is the effect of plume-scale nonlinear chemistry and microphysics on the eventual climate impact of reactive non-$CO_2$ aircraft emissions. The chemical and physical state of the atmosphere is affected by both natural variability in climate parameters, as well as by perturbations to atmospheric chemistry due to anthropogenic emissions. In the case of aircraft operating in the UTLS, the atmospheric state (and hence the chemical fate of reactive aircraft emissions) is influenced by both the background chemical composition of the surrounding air and the elevated concentrations of emissions within the aircraft exhaust plume. The sensitivity of the atmospheric response to emissions is heightened at these altitudes compared to at surface level, due to reasons elaborated on in Section 5.1. Climate-optimal aircraft routing and formation flight can both be utilised to minimise time spent flying through atmospheric regions that are unfavourable in terms of climate impact.

Real world implementation of non-carbon based operational mitigation concepts does however require: knowledge of aviation fuel consumption, plume dispersion characteristics, and monitoring and forecasting of chemical and meteorological parameters along potential routes. Moreover, this information must be obtained to a sufficiently high-resolution and within a relatively short timespan, so as to capture adequate variability in climate parameters to make an informed and timely decision on routing. As a result, modelling assumptions will inevitably need to be made, to balance computational run time with accuracy. In particular, the incorporation of plume-scale effects into large-scale climate models is likely to bring about issues with resolving data between the two resolutions. Another key issue limiting feasibility at the current time is that of air traffic management. Further work should focus on these key challenges and work towards improving scientific understanding of the non-$CO_2$ radiative forcing multiplier. Achieving a general scientific consensus on this non-$CO_2$ multiplier would increase confidence amongst policymakers to include such effects in aviation climate policy, and hence accelerate the widespread adoption of interim mitigation approaches such as climate-optimal aircraft routing and formation flight.

**Author Contributions:** Conceptualization, K.N.T.; writing—original draft preparation, K.N.T.; writing—review and editing, K.N.T., M.A.H.K., S.B., M.H.L. and D.E.S.; supervision, S.B., M.H.L. and D.E.S.; project administration, K.N.T.; funding acquisition, S.B. All authors have read and agreed to the published version of the manuscript.

**Funding:** This research was funded by the Engineering and Physical Sciences Research Council (EPSRC) as part of a Doctoral Training Partnership (DTP). M.A.H.K. was supported through a Natural Environment Research Council (NERC) Discipline Hopping for Environmental Solutions grant.

**Acknowledgments:** We would like to personally thank Stephen Roome for his invaluable technical support throughout and Thibaud M. Fritz (MIT) for his cooperation when discussing the model APCEMM.

**Conflicts of Interest:** The authors declare no conflict of interest.

## Abbreviations

The following abbreviations are used in this manuscript:

| | |
|---|---|
| PM | Particulate matter |
| ID | Instantaneous dispersion |
| EI | Emission index |
| SLS | Sea level static |
| HC | Unburnt hydrocarbons |
| VOC | Volatile organic compound |
| BADA | Base of Aircraft DAta |
| AEDT | Aviation Environment Design Tool |
| ICAO | International Civil Aviation Organisation |
| LTO | Landing and take-off cycle |
| ISA | International Standard Atmosphere |
| BFFM2 | Boeing fuel flow method 2 |
| FAR | Fuel-air ratio |
| NMHC | Non-methane hydrocarbons |
| SP | Single Plume |
| MP | Multi-layered Plume |
| APCEMM | Aircraft Plume Chemistry, Emissions, and Microphysics Model |
| LES | Large eddy simulation |
| ATM | Air traffic management |
| ATC | Air traffic control |
| NAFC | North Atlantic flight corridor |
| UTC | Universal Time Coordinated |
| POLINAT | Pollution from Aircraft Emissions in the North Atlantic Flight Corridor |
| SONEX | Subsonic Assessment Ozone and Nitrogen Oxide Experiment |
| CCN | Cloud condensation nuclei |
| SW | Short-wave |
| LW | Long-wave |
| RF | Radiative forcing |
| ERF | Effective radiative forcing |
| UTLS | Upper Troposphere and Lower Stratosphere |
| RH | Relative humidity |
| ISSR | Ice-supersaturated region |
| SWV | Stratospheric water vapour |
| GWP | Global Warming Potential |
| GTP | Global Temperature Potential |
| GCM | General circulation model |
| CTM | Chemistry-transport model |
| ECHAM | European Centre for HAmburg Model |
| CAM5 | Community Atmosphere Model 5 |
| CRI | Common Representative Intermediates |
| US EPA | United States Environmental Protection Agency |
| SZA | Solar zenith angle |
| EEI | Effective emission index |
| ECF | Effective conversion factor |
| ERR | Effective reaction rate |
| CCMod | Contrail cirrus module |
| IATA | International Air Transport Association |
| SAF | Sustainable aviation fuel |
| CORSIA | Carbon Offsetting and Reduction Scheme for International Aviation |
| aCCF | Algorithmic climate change function |
| MUAC | Maastricht Upper Area Control |
| EU ETS | European Union emissions trading scheme |
| FAA | Federal Aviation Administration |
| NASA | National Aeronautics and Space Administration |
| NCAR | National Centre for Atmospheric research |

AIAA    American Institute for Aeronautics and Astronautics
IEEE    Institute of Electrical and Electronic Engineers

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
