# Peer review of "Aircraft Emissions, Their Plume-Scale Effects, and the Spatio-Temporal Sensitivity of the Atmospheric Response: A Review"

_aerospace, doi:10.3390/aerospace9070355_

Round 1

Reviewer 1 Report

This is a nice piece of research which in my opinion is worth publishing. 

This is a task which typically precedes aircraft emission modelling, because you need to know what kind of aircraft you are dealing with in the future (and how many) and this has a pretty big influence on results. There is also a tendency to employing larger aircraft and flying routes nonstop with more efficient aircraft like e.g. the Boeing 787, which were previously not served by a nonstop service. These are important topics for aircraft emission modelling and climate impact, but I think that would be too much to be included in the paper, which is already very long. There is a section on aircraft routing, which is good, but there is much more to be said on the number and types of aircraft that underly these calculations.

Author Response

Dear Reviewer 1,

Thank you very much for your review response. I am very pleased that you rated the paper so highly and appreciate your comments.

I fully take onboard your comment about providing reference to current and future aviation best practices and equipment used. This is something which I also think I could have expanded on more, provided the paper wasn't already so long!

I do however, have future work planned, to carry out air traffic data analysis using the Opensky dataset. This will include observation of typical air traffic patterns of the current day global fleet, with the objective of locating airspace regions which have the highest air traffic density all year round (e.g. in flight corridors or over continental mainland areas). I hope this serves as an adequate response to your review, and that you can also agree that there will be no need to further expand on what has been written in this review, given that I plan to work on this in a future paper.

Best regards,

Kieran

Reviewer 2 Report

Topic of the paper is interesting - Authors tried to collect and make literature review about aviation’s non-CO2 climate impact.

Structure of the paper was explained by Authors in Introduction section. In my opinion it is correct. Review contains above 190 references.

I have one general comment which shoulkd be improved in the paper - figures:

I understand that reviewed manusctript type is review, however Autgohs should prepare own Figures. In other case Authors have to collect agreements from Figures owners to publish them in the paper.

Author Response

Dear Reviewer 2,

Thank you for your review response. I understand your concerns about figure preparation, and the need to produce original figures. However, seeing as this is a literature review I don’t think it will be appropriate to reproduce figures that I am referring to for the purpose of discussing their results. Instead, I am taking the necessary steps to obtain copyright permissions for all relevant figures.

Also, if you could please provide further information regarding your low ratings in response to the following questions “Is the work a significant contribution to the field?” and “Is the work scientifically sound and not misleading?”, that would be very much appreciated.

Best regards,

Kieran

Reviewer 3 Report

During reading, I have noticed only one mistake - Call on Figure 19 is missing in the paper text. This is the shortest review I have ever made. Great Review paper, my congratulations to the Authors.

Author Response

Dear reviewer 3,

Thank you very much for your review response. I really appreciate the feedback and have taken the necessary steps to answer the request regarding the call for fig 19. Please see manuscript for this revision.

Best regards,

Kieran